# SAMPO: Scale-wise Autoregression with Motion PrOmpt for Generative World Models

**Sen Wang**[1] **Jingyi Tian**[1] **Le Wang**[1,✉] **Zhimin Liao**[1] **Jiayi Li**[1] **Huaiyi Dong**[1]
**Kun Xia**[1] **Sanping Zhou**[1] **Wei Tang**[2] **Hua Gang**[3]

[1]National Key Laboratory of Human-Machine Hybrid Augmented Intelligence,
National Engineering Research Center for Visual Information and Applications,
Institute of Artificial Intelligence and Robotics, Xi'an Jiaotong University
[2]University of Illinois at Chicago [3]Amazon.com, Inc.

## Abstract

World models allow agents to simulate the consequences of actions in imagined environments for planning, control, and long-horizon decision-making. However, existing autoregressive world models struggle with visually coherent predictions due to disrupted spatial structure, inefficient decoding, and inadequate motion modeling. In response, we propose **S**cale-wise **A**utoregression with **M**otion **PrO**mpt (**SAMPO**), a hybrid framework that combines visual autoregressive modeling for intra-frame generation with causal modeling for next-frame generation. Specifically, SAMPO integrates temporal causal decoding with bidirectional spatial attention, which preserves spatial locality and supports parallel decoding within each scale. This design significantly enhances both temporal consistency and rollout efficiency. To further improve dynamic scene understanding, we devise an asymmetric multi-scale tokenizer that preserves spatial details in observed frames and extracts compact dynamic representations for future frames, optimizing both memory usage and model performance. Additionally, we introduce a trajectory-aware motion prompt module that injects spatiotemporal cues about object and robot trajectories, focusing attention on dynamic regions and improving temporal consistency and physical realism. Extensive experiments show that SAMPO achieves competitive performance in action-conditioned video prediction and model-based control, improving generation quality with $4.4\times$ faster inference. We also evaluate SAMPO's zero-shot generalization and scaling behavior, demonstrating its ability to generalize to unseen tasks and benefit from larger model sizes.

## 1 Introduction

Building a world model that can simulate the physical environment and respond to the actions of agents is a central challenge on the path to artificial general intelligence (AGI) [2, 11, 77, 33, 52, 72, 73]. Recently, video generation has been integrated into world models, enabling models to generate future frames based on agent actions, simulating dynamic environments and making it possible for agents to anticipate outcomes and make informed decisions [24, 7, 50, 4]. Despite growing progress, **designing a world model that is simultaneously high-fidelity, temporally consistent, and efficiently scalable remains an open problem.**

Prior works have advanced video-based world models by formulating future prediction as an action-conditioned generation problem. These approaches can be categorized into three major families based on their generative paradigms: masked modeling, diffusion-based models and autoregressive models.

---

✉Corresponding author.

39th Conference on Neural Information Processing Systems (NeurIPS 2025).

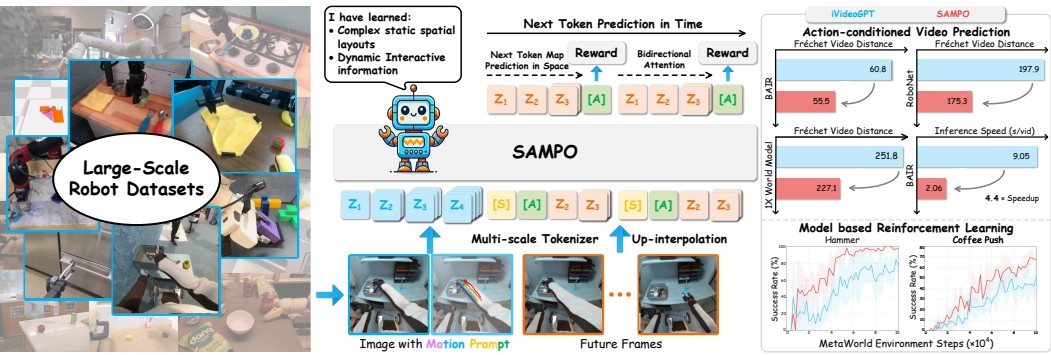

Figure 1: **SAMPO Overview.** SAMPO is a scale-wise autoregressive world model for video prediction and robotic control. It models temporal dynamics through frame-wise causal generation, while capturing spatial structure via multi-scale tokenization and coarse-to-fine prediction. A trajectory-aware motion prompt further enhances spatiotemporal grounding. SAMPO supports high-fidelity, action-conditioned rollouts for visual planning and model-based reinforcement learning.

Masked modeling [75, 8, 19, 68, 51] achieves efficient pretraining by reconstructing missing patches, yet often sacrifices temporal consistency and causality due to its localized objective. Diffusion-based models [56, 7, 26, 65] produce high-fidelity video via iterative denoising, but suffer from slow inference and limited interactivity. In contrast, autoregressive models [38, 77, 60, 62] generate tokens sequentially, preserving causal structure and supporting in-context prediction [10], making them better aligned with the requirements of world models, where accurate, interactive and temporally consistent forecasting is essential.

Despite their success in building world models [60], current autoregressive approaches still face the following limitations: 1) Structural degradation due to raster-scan flattening [45, 54, 32]. Flattening disrupts the spatial locality of video frames, hindering the model's ability to capture long-range dependencies across space, leading to physically implausible generation, such as object disappearances or blurred manipulators (Fig. 4). 2) Slow and error-prone next-token prediction leads to inefficiency and the accumulation of errors during generation [5, 41]. 3) Insufficient modeling of salient motion and interactions, which diminishes the physical realism and smoothness of dynamic scenes [74].

To address these challenges, we propose **SAMPO**, a scale-wise autoregressive framework that combines bidirectional spatial attention within frames and causal temporal modeling across time. SAMPO introduces a new autoregressive formulation tailored for world models, combining next-scale spatial prediction [45] with temporal causal generation, thereby unifying spatial coherence and temporal consistency under a scalable architecture. Specifically, the model autoregresses over time while generating each frame's token maps in a coarse-to-fine manner, progressively from low to high resolution with parallel token generation within each scale. Compared to raster-scan flattening, which disrupts spatial continuity and object boundaries [62, 32], our hierarchical token generation effectively preserves spatial locality and structural coherence within each frame and supports scalable and efficient generation across resolutions.

To further balance spatial detail and dynamic modeling [58, 60], we devise an asymmetric multi-scale tokenizer based on vector quantization [14, 48]. Observed frames are densely tokenized to preserve static background and contextual information, while future frames use sparse tokenization to emphasize dynamic changes and reduce redundancy [18]. As shown in Fig. 1, this design improves inference speed while maintaining visual fidelity. Notably, this formulation supports token-level integration of visual inputs and agent actions. With autoregressive scalability, SAMPO can be pretrained on large-scale robot datasets [37], enabling generalizable and control-centric world models across diverse tasks and settings.

While improving visual fidelity is a desirable goal, the core objective of an interactive world model is to accurately predict future states in response to agent actions [46, 33]. Existing approaches often struggle to model meaningful dynamic interactions [60, 58, 62], particularly in environments dominated by static or quasi-static frames [13, 9], resulting in blurred or inconsistent object interactions. To address this limitation, we introduce a trajectory-aware motion prompt module that provides spatiotemporal cues about object and robot trajectories within the observed frames [31, 30]. These motion prompts serve as dynamic priors, guiding the model's attention toward interaction-relevant

regions, such as robotic arms and manipulated objects. By explicitly conditioning on motion trajectories, SAMPO improves its capability to model object-agent interactions, maintain temporal consistency and capture underlying physical dependencies.

In summary, the main contributions of this study can be summarized as follows:

1. We propose SAMPO, a scale-wise autoregressive framework that combines temporal causal modeling with coarse-to-fine visual autoregression and an asymmetric multi-scale VQ tokenizer, preserving spatial locality while significantly improving generation efficiency.

2. We introduce a trajectory-aware motion prompt module that provides explicit spatiotemporal priors over robot and object trajectories, enhancing the model's ability to capture dynamic interactions and physical dependencies in complex manipulation tasks.

3. Extensive experiments demonstrate that SAMPO outperforms existing state-of-the-art methods in terms of video quality, motion modeling accuracy, and robot control performance, offering a new insight for scalable and structurally coherent world model design.

## 2 Related Work

### 2.1 Generative Models for World Modeling

**World Modeling as an Embodied Simulator.** World models have emerged as a fundamental paradigm for enabling agents to reason about and interact with complex environments. Broadly, world models serve two complementary purposes: constructing internal representations that abstract the external world and predicting its future evolution to guide decision-making [20, 33, 11]. Early works emphasized building compact, latent models that capture essential environmental dynamics, supporting tasks such as planning and policy learning in model-based reinforcement learning [21, 28]. Recent advances in video generation and large multimodal models have shifted attention toward direct pixel-level predictions of future world states [77, 7, 8, 26], providing richer supervision and expanding the applicability of world models to diverse tasks, from robotic manipulation to embodied social simulation. Beyond pixel prediction, a critical evolution in world models lies in supporting interactivity and control [60]. **An effective world model should not only generate visually plausible futures but also simulate the consequences of agent actions and respond with feedback.** This capability enables agents to interact with imagined environments in a closed-loop manner — testing actions, observing outcomes, and refining strategies accordingly [6, 11]. Such interactive modeling is essential for decision-making tasks that require dynamic adaptation, from robot manipulation to embodied reasoning. In this work, we focus on advancing interactive world models toward efficient, structurally coherent visual dynamics prediction, integrating spatial structure priors with scalable autoregressive architectures.

**Visual Autoregressive Modeling.** VAR introduces a new generation paradigm that redefines autoregressive learning as next-scale prediction, enabling transformers to better capture visual distributions [45]. By replacing raster-scan ordering with multi-scale token map prediction, VAR preserves spatial locality, reduces sequential dependency, and enables efficient parallel decoding within each scale. Inspired by these insights, coarse-to-fine multi-scale generation has begun to influence a broad range of fields, including high-resolution image synthesis[23], 3D generation [70], multimodal LLM [78, 79] and robotic manipulation [17]. However, these methods have yet to explore integrating VAR into intra-frame generation and still rely on suboptimal raster-scan ordering in image generation.

### 2.2 Motion Prompt for Visual Dynamics Modeling

Visual prompt have emerged as a lightweight alternative to architectural changes for guiding multimodal models [43, 64, 63, 67, 61]. Early efforts introduce coarse overlays or fine-grained masks to input images, steering large vision–language models toward target regions [43, 64, 67, 53]. While effective for object localization [63, 61], these prompts encode only static spatial cues and fail to capture object motion, limiting their suitability for world model learning and control. To address this limitation, recent works have introduced motion-aware prompting techniques that explicitly encode spatio-temporal dynamics [16, 74, 34, 59]. Motion prompting [16] controls video diffusion models using sparse or dense motion tracks, enabling realistic and controllable object and camera dynamics. TraceVLA [74] overlays tracked trajectories [31] as visual prompts to inject temporal context into

vision–language action models without architectural changes. Complementarily, MoVideo [34] integrates optical flow and depth features to enhance motion fidelity and temporal consistency in video generation. These methods demonstrate that motion-aware prompt can enhance dynamic fidelity without compromising efficiency. However, they often target generative tasks or depend on offline trajectories. In contrast, we propose an online motion prompting scheme that integrates with interactive world models for efficient and physically constrained visual control.

# 3 Method

In this section, we elaborate on the proposed SAMPO, a scale-wise autoregressive world model that integrates temporally causal modeling with bidirectional spatial attention in each frame. We commence with a brief background on visual autoregressive modeling and formulate the problem.

## 3.1 Preliminaries and Problem Statement

**Next-scale prediction.** VAR [45] introduces a novel generation paradigm that predicts images hierarchically from coarse to fine token maps. Instead of autoregressively generating a flattened raster-scan sequence, VAR decomposes an image into multi-scale token maps and models generation at each spatial scale, conditioned on all previous scales. Formally, given hierarchical token maps $\{z^{(1)}, z^{(2)}, \ldots, z^{(L)}\}$, where each token map $z^{(l)} \in \mathbb{Z}^{H_l \times W_l}$ from low to high resolution, the generation objective can be factorized as:

$$p(z^{(1)}, \ldots, z^{(L)}) = \prod_{l=1}^{L} p(z^{(l)} \mid z^{(1)}, \ldots, z^{(l-1)}), \tag{1}$$

where $z^{(l)}$ denotes the token map at scale $l$. Each finer scale prediction is conditional on all previously generated coarser token maps, enabling coherent and efficient spatial modeling.

This coarse-to-fine framework, while originally developed for images, preserves spatial locality and enables parallel decoding. When combined with frame-wise causal modeling, it naturally extends to spatiotemporal modeling and forms the basis of SAMPO for the structured world model.

**World Model Formulation.** We formulate world models as an interactive video prediction problem [60], where the model simulates future observations and rewards conditioned on past observations and actions, which can be formalized as a partially observable Markov decision process (POMDP), defined as: $\mathcal{M} = (\mathcal{S}, \mathcal{O}, \phi, \mathcal{A}, p, r, \gamma)$. At each timestep $t$, the agent receives a partial observation $o_t \in \mathcal{O}$, takes an action $a_t \in \mathcal{A}$, and transitions to a new latent state $s_{t+1} \sim p(s_{t+1} \mid s_t, a_t)$, receiving a reward $r_t = r(s_t, a_t)$. The objective is to learn a policy $\pi(a_t \mid o_{1:t})$ that maximizes the expected discounted return. To support this, a world model approximates the environment's transition dynamics by learning the predictive distribution: $p(o_{t+1}, r_{t+1} \mid o_{1:t}, a_{1:t})$.

## 3.2 Scale-wise Visual Autoregressive for World Models

**Hybrid Autoregressive Architecture.** We propose SAMPO, a scale-wise visual autoregressive architecture for world modeling over multimodal inputs, which unifies temporal and spatial generation through a coarse-to-fine decoding scheme. This design enables our world model to preserve both temporal causality across frames and spatial semantic coherence within each frame. Our experiments demonstrate that the hybrid architecture improves generation quality while also accelerating inference.

Specifically, given an input sequence of observation frames $V = \{f_t \in \mathbb{R}^{H \times W \times 3}\}_{t=1}^{T}$, we first discretize each frame $f_t$ into a hierarchy of multi-scale token maps using vector-quantized tokenizers [14, 48], yielding $\{z_t^{(l)} \in \mathbb{Z}^{H_l \times W_l} \mid l = 1, \ldots, L\}$, where $L$ denotes the total number of spatial scales. We then adopt a hybrid decoding scheme, which is autoregressive across frames (temporal) while generating tokens in a coarse-to-fine manner within each frame (spatial). The hybrid autoregressive likelihood is formulated as:

$$p(\{z_t^{(l)} \mid a_{<T}\}) = \prod_{t=1}^{T} \prod_{l=1}^{L} p\left(z_t^{(l)} \mid z_{<t}^{(*)}, z_t^{(<l)}, a_{<t}\right), \tag{2}$$

where $z_{<t}^{(*)}$ denotes all token maps from previous frames, $z_t^{(<l)}$ represents coarser-scale maps already decoded within the current frame, and $a_{<t}$ is the action sequence from the initial to the current time

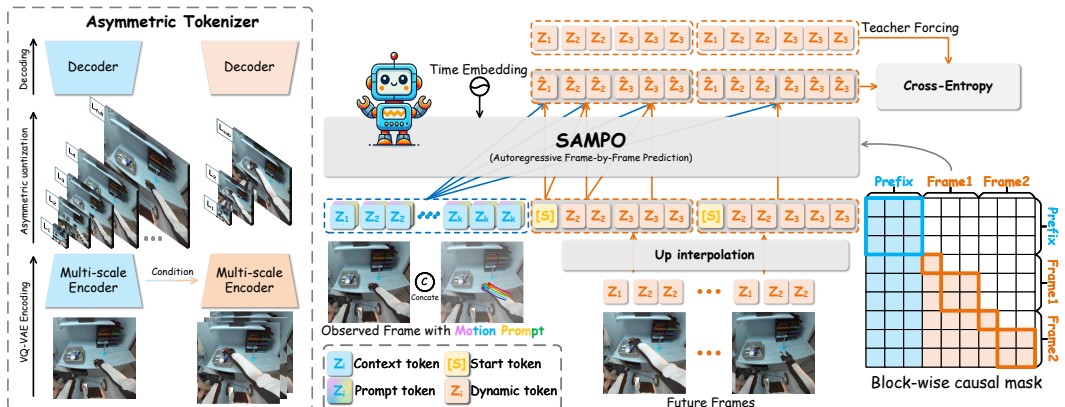

Figure 2: **The overall framework of SAMPO.** The observed and future frames are discretized by a multi-scale tokenizer to obtain dense and sparse token maps, which are then autoregressively predicted across time, while following a coarse-to-fine decoding order within each frame. Motion prompts extracted from observed frames are injected alongside visual tokens to guide dynamic modeling.

step. Actions are integrated by linear projection and added to the start token embedding. At the $k$-th generation step of the $t$-th frame, the observed frames are encoded into compact features along with previously decoded tokens, forming the prefix for generating the next-scale tokens $z_t^{(l)}$. All tokens for the current step are generated in parallel, using a block-wise causal mask to ensure each token attends only to the prefix. During inference, we employ KV-caching [15] and no mask is needed.

**Asymmetric Multi-scale Tokenizer.** In robot-centric world modeling, observed frames (*e.g.*, those acquired before taking actions) and future frames (*e.g.*, those imagined during planning) exhibit distinct characteristics [58, 60]. Observed frames typically contain complex static spatial layouts, sensor noise, and rich contextual cues. In contrast, future frames primarily reflect sparse motion involving the robotic arms and manipulated objects, while most background remains static, assuming a relatively stable camera viewpoint without significant egomotion.

To better align with this asymmetry, we propose an asymmetric multi-scale tokenizer. Observed frames ($t \leq T_0$) are tokenized across all spatial scales $l \in \{1, \ldots, L_{full}\}$, yielding fine-grained token maps with dimensions $z^{(l)} \in \mathbb{Z}^{H_l \times W_l}$ at each scale. While for future frames ($t > T_0$), we select only a sparse subset of coarser scales $l \in \{1, \ldots, L_{sub}\}$ with $L_{sub} < L_{full}$, reducing token redundancy and focusing modeling capacity on dynamic regions. The encoded token map is defined as:

$$z_t^{(l)} = \begin{cases} \mathcal{T}_l(f_t), & \text{if } t \leq T_0, \ l \in \{1, \ldots L_{full}\} \\ \mathcal{T}_l(f_t) + CrossAttn(f_t, z_{1:T_0}), & \text{if } t > T_0, \ l \in \{1, \ldots L_{sub}\} \end{cases}, \quad (3)$$

where $\mathcal{T}_l(\cdot)$ denotes the scale-specific tokenizer at scales $l$. For future frames ($t > T_0$), cross-attention incorporates information from observed frames. This asymmetry leads to efficient and structured representation by enhancing scale-aware attention during generation and disentangling static background priors from dynamic foreground variations.

### 3.3 Trajectory-aware Motion Prompt

While next scale autoregression improves spatial coherence, it remains insufficient for dynamic understanding, especially under static or quasi-static training distributions [13, 9]. To mitigate this, we incorporate explicit trajectory-aware motion prompts that guide the model to focus on dynamically relevant regions. We extract motion prompts using CoTracker3 [30], a point-tracking model. Specifically, we adopt the `scaled_online`[1] variant for efficient and robust trajectory extraction.

Following the definition of $f_t$ in Sec. 3.2, we uniformly sample a regular grid of query points on the first frame $f_1$ with a predefined grid size $G \times G$, resulting in $N = G^2$ initial points $\{(x_i^{(1)}, y_i^{(1)})\}_{i=1}^{N}$.

---

[1] A lightweight variant fine-tuned on 15k real-world videos via pseudo-labeling, showing significant improvements in robotic tracking benchmarks such as RoboTAP [49].

These query points are tracked across frames by CoTracker3, generating raw trajectories:

$$\mathcal{P}_i = \{(x_i^{(t)}, y_i^{(t)})\}_{t=1}^{T}, \quad \text{for } i = 1, \ldots, N, \tag{4}$$

where $(x_i^{(t)}, y_i^{(t)})$ denotes the predicted 2D location of point $i$ at frame $t$. For world modeling in robotic manipulation, we extract trajectories to focus on the dynamics of the robot arms and manipulated objects, rather than static backgrounds or noisy artifacts. To this end, we filter raw CoTracker3 outputs based on two criteria. First, trajectories with low average tracking confidence $c_i < \tau_c$ are discarded. Second, we compute the displacement over a short temporal window $\Delta t = 4$ frames to identify static points:

$$d_i^{(\Delta t)} = \|(x_i^{(t+\Delta t)}, y_i^{(t+\Delta t)}) - (x_i^{(t)}, y_i^{(t)})\|_2, \tag{5}$$

and discard trajectories where $d_i^{(\Delta t)}$ remains below 2% of the image diagonal throughout the sequence. The resulting dynamic trajectories serve as motion prompts, enabling spatiotemporal reasoning about interactive agents and objects. The retained dynamic trajectories are overlaid onto the original observation to form the motion prompt (see Fig. 3). Following [74], we concatenate the motion prompt with the observation, separated by a special token, to construct a dual-branch input. This design enhances the spatiotemporal grounding of the model and improves its ability to simulate physically plausible interactions.

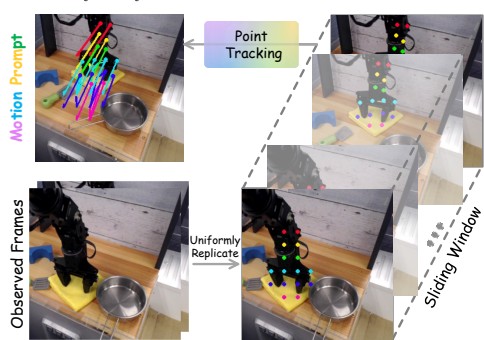

Figure 3: **Motion Prompt generation.**

## 3.4 Training Objectives

SAMPO is trained to autoregressively predict future token maps over both temporal and spatial scales. Let $\hat{z}_t^{(l)} \in \mathbb{R}^{H_l \times W_l \times V}$ denote the predicted logits at scales $l$, where $V$ represents the size of the codebook, and $z_t^{(l)} \in \mathbb{Z}^{H_l \times W_l}$ are the corresponding ground-truth token indices. The training objective is a multi-scale cross-entropy loss over future frames:

$$\mathcal{L}_{\text{CE}}(\theta) = \sum_{l=1}^{L} \lambda_l \cdot \mathbb{E}_{t > T_0} \left[ \frac{1}{H_l W_l} \sum_{i,j} -\log p\left(\hat{z}_t^{(l)}(i,j) = z_t^{(l)}(i,j)\right) \right] \tag{6}$$

where $\lambda_l$ is a scale-specific weight (detailed in Appendix A.2), and $T_0$ denotes the final observed frame. The loss is computed only on future frames, encouraging prediction conditioned on context. **Action-conditioned prediction with reward.** For control-centric applications (*e.g.*, model-based RL), the objective can be optionally extended with auxiliary heads, including reward prediction or trajectory decoding. For example, a linear head can be applied to the last token's hidden state in each frame, using a MSE loss for reward prediction [60]. This modification enables more effective task-relevant learning, improving performance in control tasks [35].

## 3.5 Implementation Details

**Asymmetric Tokenizer.** We design an asymmetric multi-scale tokenizer built on VQGAN [14], which independently encodes observed and future frames using separate codebooks of size 8192. Both share a CNN backbone but their parameters are independently updated. Our tokenizer is pretrained on Open X-Embodiment [37] with reconstruction and commitment loss.

**Transformer.** SAMPO follows the decoder-only architecture [40] implemented in VAR [45]. To maintain temporal consistency, we introduce a start token [S] at the beginning of each frame, which segments the frame and enables autoregressive prediction with teacher forcing. Additionally, we apply fixed 1D sine-cosine embeddings to encode temporal positions, following standard practice in visual Transformers [39, 12, 25]. By default, our Transformer pretrained on Open X-Embodiment [37] has 16 blocks and a width of 1024, which we refer to as the Base size or -B (~353M parameters).

**Motion Prompt.** We apply a dropout mechanism to randomly omit motion prompts, avoiding the limitation of using a single observation frame and enhancing robustness across both prompted and unprompted conditions. During inference, when the number of observed frames is shorter than the required window size [30], frames are uniformly replicated to satisfy the temporal input constraint. Full training recipe including model architecture and data preprocessing is provided in Appendix A.

Table 1: **Video prediction results on BAIR and RoboNet datasets.** "-" indicates values not reported. We report Fréchet Video Distance (FVD) [47] as the primary metric, complemented by PSNR [27], SSIM [55], and LPIPS [71] for perceptual quality assessment. LPIPS and SSIM are scaled by 100.

| **BAIR** [13] | FVD↓ | PSNR↑ | SSIM↑ | LPIPS↓ | **RoboNet** [9] | FVD↓ | PSNR↑ | SSIM↑ | LPIPS↓ |
|---|---|---|---|---|---|---|---|---|---|
| *action-free & 64×64 resolution* | | | | | *action-conditioned & 64×64 resolution* | | | | |
| MaskViT [19] | 93.7 | - | - | - | MaskViT [19] | 133.5 | 23.2 | 80.5 | 4.2 |
| FitVid [4] | 93.6 | - | - | - | FitVid [4] | 62.5 | 28.2 | 89.3 | **2.4** |
| MCVD [51] | 89.5 | 16.9 | 78.0 | - | SVG [50] | 123.2 | 23.9 | 87.8 | 6.0 |
| MAGVIT [68] | 62.0 | 19.3 | 78.7 | 12.3 | GHVAE [57] | 95.2 | 24.7 | 89.1 | 3.6 |
| iVideoGPT [60] | 75.0 | 20.4 | 82.3 | 9.5 | iVideoGPT [60] | 63.2 | 27.8 | 90.6 | 4.9 |
| SAMPO | **65.7** | **22.3** | **86.7** | **8.4** | SAMPO | **57.1** | **29.3** | **94.1** | 3.3 |
| *action-conditioned & 64×64 resolution* | | | | | *action-conditioned & 256×256 resolution* | | | | |
| MaskViT [19] | 70.5 | - | - | - | MaskViT [19] | 211.7 | 20.4 | 67.1 | 17.0 |
| iVideoGPT [60] | 60.8 | 24.5 | 90.2 | 5.0 | iVideoGPT [60] | 197.9 | 23.8 | 80.8 | 14.7 |
| SAMPO | **55.5** | **26.7** | **94.7** | **3.7** | SAMPO-L | **175.3** | **25.3** | **84.7** | **12.3** |

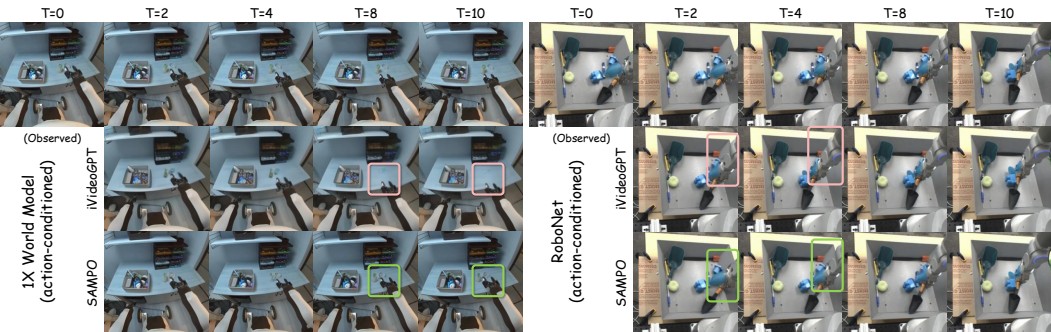

Figure 4: **Qualitative comparison of video prediction.** We compare SAMPO with iVideoGPT on the 1X World Model and RoboNet datasets under action-conditioned settings. SAMPO shows a clear advantage in modeling complex backgrounds and capturing dynamic object interactions over time.

## 4 Experiments

We conduct extensive experiments to validate the effectiveness of SAMPO across multiple settings, including action-free and action-conditioned video prediction, visual planning, and model-based reinforcement learning (MBRL). We follow standardized evaluation protocols [60] and report results on established benchmarks. Training details for each benchmark are provided in Appendix B.

### 4.1 Video Prediction Performance

**Datasets and Setup.** We evaluate on three categories of benchmarks: (i) BAIR Robot Pushing [13] for low-resolution video prediction, (ii) RoboNet [9] for large-scale action-conditioned prediction, and (iii) 1X World Model [1] for real-world human and robotic interactions in diverse indoor environments, designed for open-domain video prediction. We predict 15 future frames from 1 context frame on BAIR, 10 future frames from 2 context frames on RoboNet and 1X World Model. Action-conditioned is applied during rollouts.

In Tab. 1 and Tab. 2, SAMPO achieves the best FVD in video prediction. Qualitative comparisons, shown in Fig. 4, demonstrate significant improvements in perceptual quality and motion realism. These results highlight the effectiveness of scale-wise generation in modeling multi-scale dynamics and the role of trajectory-aware motion prompts in guiding future predictions.

Table 2: **Video prediction results on 1X.**

| **1X** [1] | FVD↓ | PSNR↑ | SSIM↑ | LPIPS↓ |
|---|---|---|---|---|
| *action-conditioned & 256×256 resolution* | | | | |
| iVideoGPT | 251.8 | 24.1 | 78.3 | 20.3 |
| SAMPO | **227.1** | **25.7** | **80.3** | **18.7** |

Table 3: **Visual planning performance in VP$^2$.** We report the success rates across 8 tasks, and the average success rate excluding Flat Block. In addition, we provide the mean and standard deviation of the success rates (in %) on average in 3 random seeds.

| Method / Task | Robosuite Push | Flat Block | Open Drawer | Open Slide | Blue Button | Green Button | Red Button | Upright Block | Avg. Success↑ |
|---|---|---|---|---|---|---|---|---|---|
| Simulator | $93.5^{\pm1.2}$ | $13.3^{\pm0.0}$ | $76.7^{\pm0.0}$ | $71.7^{\pm1.4}$ | $100.0^{\pm0.0}$ | $96.7^{\pm0.0}$ | $90.0^{\pm0.0}$ | $90.0^{\pm0.0}$ | 100.0 |
| FitVid [4] | $67.7^{\pm6.4}$ | $\mathbf{9.2}^{\pm2.8}$ | $25.3^{\pm8.2}$ | $\underline{35.3}^{\pm5.5}$ | $94.0^{\pm4.2}$ | $\underline{84.0}^{\pm5.5}$ | $58.7^{\pm5.5}$ | $51.3^{\pm2.9}$ | 65.6 |
| SVG' [50] | $79.8^{\pm3.3}$ | $2.0^{\pm1.4}$ | $16.7^{\pm8.2}$ | $\mathbf{57.3}^{\pm11.0}$ | $\underline{97.3}^{\pm2.7}$ | $81.3^{\pm5.5}$ | $76.0^{\pm9.5}$ | $\underline{48.7}^{\pm11.1}$ | $\mathbf{72.5}$ |
| MCVD [51] | $77.3^{\pm2.1}$ | $4.0^{\pm1.4}$ | $11.7^{\pm1.4}$ | $18.3^{\pm1.4}$ | $95.0^{\pm4.1}$ | $83.3^{\pm0.0}$ | $73.3^{\pm2.7}$ | $56.7^{\pm2.7}$ | 64.3 |
| MaskViT [19] | $\mathbf{82.6}^{\pm2.5}$ | $4.0^{\pm4.1}$ | $4.0^{\pm4.1}$ | $8.7^{\pm5.5}$ | $94.7^{\pm1.4}$ | $64.0^{\pm4.1}$ | $24.0^{\pm8.2}$ | $\mathbf{62.2}^{\pm9.5}$ | 52.1 |
| Struct-VRNN [36] | $55.4^{\pm4.1}$ | $4.7^{\pm5.5}$ | $2.7^{\pm4.2}$ | $12.7^{\pm6.9}$ | $86.7^{\pm4.2}$ | $68.0^{\pm9.5}$ | $30.7^{\pm4.2}$ | $33.3^{\pm2.7}$ | 44.2 |
| iVideoGPT [60] | $78.3^{\pm0.8}$ | $3.3^{\pm0.7}$ | $\underline{37.5}^{\pm1.7}$ | $16.1^{\pm2.7}$ | $95.6^{\pm2.1}$ | $82.5^{\pm3.4}$ | $\underline{92.2}^{\pm1.4}$ | $44.7^{\pm1.7}$ | 70.1 |
| SAMPO | $\underline{80.7}^{\pm1.4}$ | $\underline{5.5}^{\pm1.2}$ | $\mathbf{40.3}^{\pm2.3}$ | $18.3^{\pm3.3}$ | $\mathbf{97.3}^{\pm1.7}$ | $\mathbf{85.3}^{\pm3.3}$ | $\mathbf{94.7}^{\pm2.1}$ | $46.1^{\pm2.7}$ | $\underline{72.2}$ |

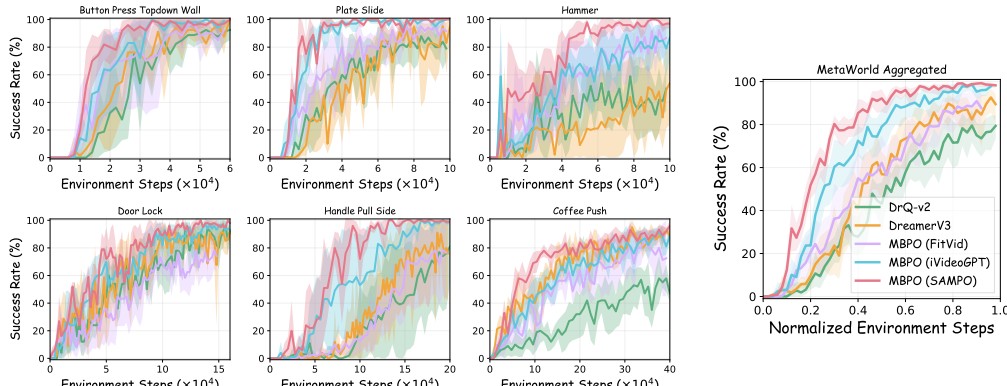

Figure 5: **Model-based RL with SAMPO.** We report the success rate (in %) across 6 tasks, along with the average success rate across all tasks and the 95% confidence interval [3] calculated from 5 random seeds. All models are pre-trained with world models [42].

## 4.2 Planning and Reinforcement Learning

**Visual Planning with Action-Conditioned Rollouts.** As perceptual metrics are not strictly correlated with control performance, we further verify the applicability of SAMPO on VP$^2$ [46], a visual planning benchmark. Each environment provides noisy, scripted interaction trajectories. Following [46, 60], we train SAMPO on 5k trajectories for Robosuite [76] and 35k for RoboDesk [29], and compare against established baselines.

As shown in Tab. 3, SAMPO achieves the best results in four tasks and the second-best results in the other two tasks. Compared to the baseline [60], the average performance is improved by 2.1%, indicating its capability to achieve high-fidelity perception while excelling in control performance.

**Model-Based Reinforcement Learning.** Beyond planning, an effective world model must facilitate efficient policy learning through interaction. In this work, we evaluate SAMPO by incorporating it into a model-based policy optimization (MBPO) framework [28], which extends the replay buffer with synthetic rollouts to train an actor-critic RL algorithm, implemented based on DrQ-v2 [66]. Experiments are conducted on six robotic manipulation tasks from the Meta-World benchmark [69].

Fig. 5 shows that SAMPO significantly accelerates policy convergence compared to iVideoGPT[60], while also improving the policy's upper bound. This enhancement is primarily driven by the temporal consistency and semantic coherence facilitated by our hybrid framework, in conjunction with the motion prompt. Moreover, SAMPO significantly outperforms the state-of-the-art MBRL method, DreamerV3 [22], regarding both sample efficiency and success rate.

## 4.3 Ablation Studies & Model Analysis

We perform ablation studies to isolate the contributions of each component. Results are in Tab. 4.

Table 4: **Abalation of SAMPO** on the action-conditioned RoboNet [9] at a resolution of 256×256. The first two rows compare the baseline AR model with SAMPO model using a hybrid VAR architecture. Later rows add enhancements to SAMPO, including motion prompt, 1D temporal position embedding, and model scaling. "Δ": improvement over the SAMPO-S model .

| Description | # Para. | Model | Motion | T. E. | FVD↓ | SSIM↑ | Δ |
|---|---|---|---|---|---|---|---|
| AR | 436M | AR [60] | ✗ | ✗ | 197.9 | 80.8 | - |
| Hybrid AR | 207M | SAMPO-S | ✗ | ✗ | 227.4 | 76.4 | (0.00, 0.00) |
| + Motion | 232M | SAMPO-S | ✓ | ✗ | 217.8 | 78.8 | (-9.6 , +2.4) |
| + Temp. Embed. | 232M | SAMPO-S | ✓ | ✓ | 193.8 | 81.5 | (-33.6, +5.1) |
| + Scale up | 353M | SAMPO-B | ✓ | ✓ | 184.1 | 83.2 | (-43.3, +6.8) |
| | 548M | SAMPO-L | ✓ | ✓ | 175.3 | 84.7 | (-52.1, +8.3) |

Table 5: **Performance and speed trade-off**. We benchmark FVD, PSNR, average success rate on $VP^2$ and inference speed using one A800 GPU with a batch size of 16 on the BAIR.

| Method | Spatial Scales | FVD↓ | PSNR↑ | Avg. Success↑ | Inference Time↓ |
|---|---|---|---|---|---|
| $AR_{138M}$ [60] | - | 60.8 | 24.5 | 70.1 | 9.05 s / vid. |
| SAMPO-S | [1, 2, 4, 8] | 80.7 | 23.1 | 72.2 | 1.61 s / vid |
| SAMPO-S | [1, 2, 4, 8, 10] | 73.1 | 24.3 | 71.3 | 3.27 s / vid |
| SAMPO-S | [1, 2, 3, 4, 5, 6] | **55.5** | **26.7** | 70.6 | 2.06 s / vid. |

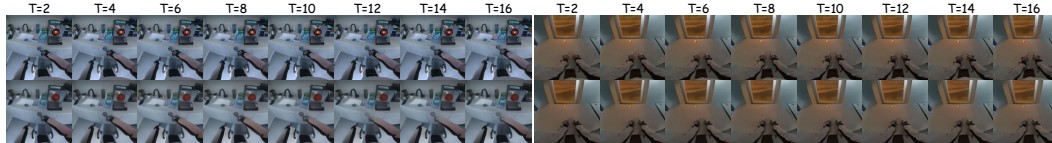

Figure 6: **Zero-shot performance.** The results show that SAMPO can effectively generalize without special design and finetuning, underscoring its potential for world models. Zoom in for a better view.

**Effectiveness of Hybrid Framework.** First, we evaluate the effectiveness of the hybrid framework, which combines scale-wise autoregressive generation with bidirectional spatial attention. Starting with the SAMPO-S model, we observe competitive performance in both FVD and SSIM, confirming its ability to generate higher-quality video. The parallel prediction of multiple tokens within each scale effectively reduces autoregressive steps, improving efficiency. In Tab. 5, SAMPO with various scales configurations significantly improve the inference speed. These demonstrate that our hybrid framework not only enhances generation quality but also accelerates inference.

**Motion Prompt and Temporal Embedding.** Both motion prompts and temporal position embedding improve temporal consistency and semantic coherence, as shown in Tab. 4. Particularly, temporal position embedding plays a crucial role in maintaining consistent temporal dynamics, resulting in significant improvements in FVD and SSIM.

**Analysis of Spatial Scale Design.** Fewer scales improve inference speed but reduce spatial fidelity. Notably, smaller inter-scale strides yield better performance under similar compute budgets, indicating that residual-based scale-wise autoregression benefits from gradual resolution refinement due to its sensitivity to abrupt scale transitions.

**Scaling Laws.** As a GPT-style world model, we explore the scaling behavior of SAMPO with 3 different sizes (depth 12, 16, 20) in Tab. 4. The observed improvements in FVD and SSIM align with the scaling laws, as larger models capture more complex temporal and spatial dependencies, improving both generation quality and consistency.

**Zero-shot Generalization.** We further assess SAMPO's zero-shot generalization by evaluating a model pretrained on the Open X-Embodiment dataset [37] and tested on the 1X World Model dataset [1] without fine-tuning. As shown in Fig. 6, SAMPO generates high-quality videos with temporally consistent dynamics, demonstrating its strong generalization ability. More zero-shot visual examples are in Appendix C.

# 5 Conclusion and Discussion

In this work, we present SAMPO, a scale-wise autoregressive world model. It integrates a hybrid autoregressive framework with an asymmetric tokenizer to perform temporal and spatial token maps generated across multiple scales, and further leverages a lightweight motion prompt module to enhance dynamic scene understanding. Owing to its hybrid architecture, SAMPO maintains both temporal consistency and spatial coherence, achieving state-of-the-art performance in video prediction and model-based RL benchmarks. However, despite its strong accuracy and scalability in both simulated and reinforcement learning environments, SAMPO still struggles with long-term modeling, leading to error accumulation over time. Future work could explore techniques for improving long-term consistency and reducing error propagation.

## Acknowledgement

This work was supported in part by the National Key Research and Development Project under Grant 2024YFB4708100, National Natural Science Foundation of China under Grants 62088102, U24A20325 and 12326608, and Key Research and Development Plan of Shaanxi Province under Grant 2024PT-ZCK-80.

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

# A Implementation Details

## A.1 Details of Network

Table A: **Hyperparameters of Asymmetric Tokenizer.**

| Tokenizer | Low-resolution | High-resolution |
|---|---|---|
| Parameters | 114M | 310M |
| Resolution | $64 \times 64$ | $256 \times 256$ |
| Obs. scale | [1, 2, 3, 4, 5, 6, 8, 10, 13, 16] | [1, 2, 3, 4, 5, 6, 8, 10, 13, 16] |
| Fut. scale | [1, 2, 3, 4, 5, 6] | [1, 2, 3, 4, 5, 6, 8, 10] |
| Embedding dim | 64 | 64 |
| Codebook size | 8192 | 8192 |
| Norm | GroupNorm | GroupNorm |
| Norm group | 32 | 32 |
| Activation | SiLU | SiLU |

Table B: **Hyperparameters of Transformer.**

| Transformer | SAMPO-S | SAMPO-B | SAMPO-L |
|---|---|---|---|
| Parameters | 207M | 328M | 523M |
| Layers | 12 | 16 | 20 |
| Heads | 12 | 16 | 20 |
| Hidden dim | 768 | 1024 | 1280 |
| Feedforward dim | 768 | 1024 | 1024 |
| Dropout | 0.05 | 0.067 | 0.083 |
| Norm | LayerNorm | LayerNorm | LayerNorm |
| Activation | GELU | GELU | GELU |

Table C: **Hyperparameters for Training.**

| Tokenizer/Transformer | Low-resolution$_{(64\times64)}$ | | | | High-resolution$_{(256\times256)}$ | | |
|---|---|---|---|---|---|---|---|
| Config | Pre-train [37] | BAIR [13] | RoboNet [9] | VP2 [46] | Pre-train [37] | RoboNet [9] | 1X WM [1] |
| GPU Device | | | | $8 \times$ A800 | | | |
| GPU days | 20 / 24 | 3 / 2 | 10 / 12 | 4 / 3 | 20 / 11 | 10 / 27 | 10 / 27 |
| Training iteration | 1M / 1M | 0.2M / 0.1M | 0.6M / 0.6M | 0.2M / 0.2M | 0.3M / 0.4M | 0.15M / 0.5M | 0.15M / 0.5M |
| Batch size | 128 / 128 | 128 / 128 | 128 / 128 | 64 / 16 | 32 / 32 | 32 / 32 | 32 / 32 |
| Sequence length | 16 / 16 | 16 / 16 | 12 / 12 | 12 / 12 | 16 / 16 | 12 / 12 | 12 / 12 |
| Context frames | 2 / 2 | 1 / 1 | 2 / 2 | 2 / 2 | 2 / 2 | 2 / 2 | 2 / 2 |
| Future frames | 6 / - | 7 / - | 6 / - | 6 / - | 6 / - | 6 / - | 6 / - |
| Learning rate | $5 / 1\ 1e^{-4}$ | $1 / 1\ 1e^{-4}$ | $1 / 1\ 1e^{-4}$ | $1 / 1\ 1e^{-4}$ | $5 / 1\ 1e^{-4}$ | $1 / 1\ 1e^{-4}$ | $1 / 1\ 1e^{-4}$ |
| LR Schedule | | | | Cosine | | | |
| Weight decay | | | | 0.01 | | | |
| Grad clip | | | | 1.0 | | | |
| Warmup steps | | | | 5000 | | | |
| Optimizer | | | | AdamW | | | |
| Gradient moment | | | | (0.9, 0.999) | | | |
| Weight decay | | | | 0.0 / 0.01 | | | |
| Mixed precision | | | | bf16 | | | |
| Motion prompt dropout ratio | | | | 0.5 | | | |
| Sampling top-k | | | | 100 | | | |
| Sampling top-p | | | | 1.0 | | | |

**Tokenizer.** In the proposed asymmetric multi-scale tokenizer, we adopt a hierarchical design to balance spatial detail preservation and computational efficiency [45]. As detailed in Tab. A, for observed frames ($t \leq T_0$), dense tokenization is applied across all spatial scales $L_{full} = 10$. For future frames ($t > T_0$), a sparse subset of coarser scales $L_{full} = 6$ is used, focusing on dynamic regions while minimizing redundancy [18]. Both observed and future frames use a codebook size of 8192 with an embedding dimension of 64.

**Transformer.** The SAMPO transformer utilizes a scalable decoder-only architecture, inspired by GPT-2, to efficiently model spatiotemporal dynamics. Unlike iVideoGPT [60], which relies on specialized tokens for frame segmentation, we initialize each frame with a single start token [S]. This token serves a dual function: it marks the beginning of intra-frame autoregressive generation and naturally defines the temporal boundaries between consecutive frames. Fixed 1D sine-cosine positional embeddings are applied to encode temporal dynamics, following standard practices in vision Transformers [39, 12, 25, 44]. We design a set of models with different sizes, as illustrated in Tab. B. Model scaling adheres to a linear relationship between depth $d$, width $w$, head count $h$, and dropout rate $dr$:

$$w = 64d, \qquad h = d, \qquad dr = 0.1 \cdot d/24. \tag{7}$$

## A.2 Details of Training

**Training setup.** Tab. C summarizes the hyperparameters used in SAMPO across datasets. Training proceeds via uniform segment sampling with dataset-specific step sizes (see Tab. D), where step lengths are tuned to match each dataset's native temporal frequency. For tokenizer training, we use the initial observed frames as context, while the transformer is trained on full-length sequences.

**Scale-specific weight.** The scale-specific weight $\lambda_l$ in Eq. (6) balances the contribution of each spatial scale to the total loss. Given a multi-scale tokenizer with patch sizes $L_{patch} = [1, ..., L_K]$, where $K = \texttt{len}(L_{patch})$ is the number of spatial scales, $\lambda_l$ is defined as:

$$\lambda_l = \frac{L_l^2}{\sum_{k=1}^{K} L_k^2} \cdot K. \tag{8}$$

where $L_l$ denotes the spatial dimension of patches at scale $l$, and $L_l^2$ denotes the number of tokens per scale (*e.g.*, a $L_l \times L_l$ patch grid contains $L_l^2$ tokens). The denominator $\sum_{k=1}^{K} L_k^2$ normalizes the token counts across all scales, ensuring that finer-grained resolutions (larger $L_l$) are not overshadowed by coarser ones (smaller $L_l$). This ensures the model prioritizes dynamic regions over static ones, which are more likely to change across frames. Such a weighted strategy is essential for capturing the underlying spatial patterns in robot-centric world modeling, leading to more effective learning.

---

**Algorithm 1** 1X Dataset Preprocessing Pipeline

---

**Require:** Dataset root paths $\mathcal{D}_{\text{train}}, \mathcal{D}_{\text{val}}$, Target frame rate $f_{\text{target}}$, Minimum segment length $T_{\min} = 51$, Clip length $T_{\text{clip}} = 30$
**Require:** Joint index mapping $\mathcal{J} : \{0, \ldots, 24\} \to \{\text{HIP\_YAW}, \ldots, \text{Angular Velocity}\}$
1: **repeat**
2:    $\mathcal{M}, \mathcal{S}, \mathcal{R}, \mathcal{V} \sim \mathcal{D}_{\text{train}}$    ▷ *Load metadata, segment indices, robot states, and video frames*
3:    $\mathcal{S}, \mathcal{R}, \mathcal{V} \leftarrow \text{Downsample}(\mathcal{S}, \mathcal{R}, \mathcal{V}, \delta f = \lfloor \frac{30}{f_{\text{target}}} \rfloor)$    ▷ *Temporal downsampling*
4:    $\mathcal{U} \leftarrow \text{UniqueSegments}(\mathcal{S})$    ▷ *Extract unique action segments*
5:    **for all** $s \in \mathcal{U}$ **do**
6:        $\tau_{\text{start}}, \tau_{\text{end}} \leftarrow \text{FindBounds}(\mathcal{S} = s)$    ▷ *Segment boundary detection*
7:        **if** $\tau_{\text{end}} - \tau_{\text{start}} < T_{\min}$ **then**
8:            **continue**    ▷ *Skip short segments*
9:        **end if**
10:        $\mathcal{V}_s \leftarrow \mathcal{V}[\tau_{\text{start}} : \tau_{\text{end}}], \mathcal{R}_s \leftarrow \mathcal{R}[\tau_{\text{start}} : \tau_{\text{end}}]$    ▷ *Sequence cropping*
11:        $\mathcal{W} \leftarrow \text{SlidingWindow}(\mathcal{V}_s, T_{\text{clip}})$    ▷ *Create temporal windows*
12:        **for all** $w \in \mathcal{W}$ **do**
13:            $\mathcal{V}_w \leftarrow \mathcal{V}_s[w : w + T_{\text{clip}}], \mathcal{R}_w \leftarrow \mathcal{R}_s[w : w + T_{\text{clip}}]$    ▷ *Windowed subsequences*
14:            **if** $|\mathcal{V}_w| < 15$ **then**
15:                **continue**    ▷ *Skip ultra-short clips*
16:            **end if**
17:            $\text{SaveNPZ}(\mathcal{V}_w, \mathcal{R}_w, \text{"train"})$    ▷ *Compressed storage*
18:        **end for**
19:    **end for**
20:    $\text{ProcessValidation}(\mathcal{D}_{\text{val}})$    ▷ *Symmetric validation processing*
21: **until** Dataset processed

---

Table D: **Detailed Dataset Mixture.** We include the detailed number of trajectories and the number of dataset sampling weight in the pretraining mixture. These include 41 dataset from Open X-Embodiment [37].

| Dataset | # Traj. | Step size | Sampling weight |
|---|---|---|---|
| **Pretrain** | | | |
| Kuka | 580392 | 3 | 8.33% |
| Language Table | 442226 | 3 | 8.33% |
| Fractal (RT-1) | 87212 | 1 | 8.33% |
| RoboNet | 82649 | 1 | 8.33% |
| BC-Z | 43264 | 3 | 8.33% |
| Bridge | 28935 | 2 | 8.33% |
| Droid | 29437 | 10 | 8.33% |
| Agent Aware Affordances | 24000 | 66.6 | 8.33% |
| ManiSkill Dataset | 21346 | 20 | 8.33% |
| Robo Set | 15603 | 5 | 7.5% |
| Functional Manipulation Benchmark | 15350 | 10 | 7.5% |
| Isaac Arnold Image | 3214 | 15 | 0.3125% |
| Stanford MaskViT | 9200 | 1 | 0.3125% |
| UIUC D3Field | 768 | 1 | 0.3125% |
| Taco Play | 3603 | 5 | 0.3125% |
| Jaco Play | 1085 | 3 | 0.3125% |
| Roboturk | 1995 | 3 | 0.3125% |
| Viola | 150 | 7 | 0.3125% |
| Toto | 1003 | 10 | 0.3125% |
| Columbia Cairlab Pusht Real | 136 | 3 | 0.3125% |
| Stanford Kuka Multimodal Dataset | 3000 | 7 | 0.3125% |
| Stanford Hydra Dataset | 570 | 3 | 0.3125% |
| Austin Buds Dataset | 50 | 7 | 0.3125% |
| NYU Franka Play Dataset | 456 | 1 | 0.3125% |
| Furniture Bench Dataset | 5100 | 3 | 0.3125% |
| UCSD Kitchen Dataset | 150 | 1 | 0.3125% |
| UCSD Pick and Place Dataset | 1355 | 1 | 0.3125% |
| Austin Sailor Dataset | 240 | 7 | 0.3125% |
| UTokyo PR2 Tabletop Manipulation | 240 | 3 | 0.3125% |
| UTokyo Xarm Pick and Place | 102 | 3 | 0.3125% |
| UTokyo Xarm Bimanual | 70 | 3 | 0.3125% |
| KAIST Nonprehensile | 201 | 3 | 0.3125% |
| DLR SARA Pour | 100 | 3 | 0.3125% |
| DLR SARA Grid | 107 | 3 | 0.3125% |
| DLR EDAN Shared Control | 104 | 3 | 0.3125% |
| ASU Table Top | 110 | 4 | 0.3125% |
| UTAustin Mutex | 1500 | 7 | 0.3125% |
| Berkeley Fanuc Manipulation | 415 | 3 | 0.3125% |
| CMU Playing with Food | 174 | 3 | 0.3125% |
| CMU Play Fusion | 576 | 2 | 0.3125% |
| CMU Stretch | 135 | 3 | 0.3125% |
| USC Cloth Sim | 684 | 10 | 0.3125% |
| Mimic Play | 323 | 10 | 0.3125% |
| **Total** | 1,407,330 | - | 100.0% |

## A.3 Details of Data

**Real Robot Dataset.** In total, we use a subset of 41 datasets in the Open X-Embodiment dataset [37], as shown in Tab. D. The datasets used in this work consist of both real-world data and simulator-generated data, providing a rich and diverse foundation for action-conditioned video prediction and model-based reinforcement learning.

**Preporcess on 1X World Model Dataset.** The 1X World Model Dataset is a large-scale multimodal dataset for training and evaluating robotic world models in dynamic environments. It includes sensory data from 1X Technologies' EVE humanoid robots performing tasks like door opening, cloth folding, and obstacle navigation. The dataset contains synchronized 512×512 RGB video frames,

25-dimensional proprioceptive state vectors, and metadata, with state vectors capturing joint angles, gripper openness, and velocities. Video frames are stored in MP4 format, with segmentation and configuration details provided in binary segment indices and JSON files.[2]

Preprocessing follows custom pipelines developed to align with iVideoGPT [60] evaluation protocols, as detailed in Algorithm 1, including parsing raw files, frame-state alignment, and task-specific normalization. This dataset facilitates standardized evaluation of temporal dependency learning, multi-modal integration, and real-world generalization for robotics.[3]

## B  Benchmark Setup

Our experiments follow the same evaluation protocols as iVideoGPT [60]. For completeness, we provide a brief introduction to the following three experiments:

**Video Prediction.** We evaluate SAMPO using four metrics: SSIM [55], PSNR [27], LPIPS [71], and FVD [47]. In line with previous work [62, 60, 4, 51, 19], we address the stochasticity of video prediction by sampling 100 future trajectories for each test video and selecting the best one for PSNR, SSIM, and LPIPS. All 100 samples are used for FVD evaluation.

**Visual Planning.** We finetuning the pretrained SAMPO on VP2 datasets and integrating it into an interface compatible with the official VP2 visual planning code. [4] For Robosuite tasks, a trajectory is deemed successful if the reward, which reflects the distance to the goal, falls below 0.05. In contrast, for Robodesk tasks, success is defined by a reward value below -0.5, with the environment returning either 0 or -1, where -1 signifies success.

**Visual Model-based RL.** For Model Rollout, the initial rollout batch size is set to 640, with an interval of 200 environment steps, a batch size of 32, and a horizon of 10. For Model Training, the initial training steps are 1000. The tokenizer training interval is 40 environment steps, while the transformer training interval is 10 environment steps. The batch size is 16, with a sequence length of 12, context frames set to 2, and 5 sampled future frames for tokenization. The learning rate is $1e^{-4}$, with no weight decay, and the optimizer used is Adam. The model-based RL real data ratio is 0.5.

## C  Additional Visualization

In this section, we present additional qualitative results of SAMPO across various datasets to complement the main text. We showcase video prediction results in Fig. 7, 8, 9, 10, 11, 12; zero-shot performance in Fig. 13; the illustration of motion prompt in Fig. 14; and the a visual comparison with the iVideoGPT [60] in Fig. 15.

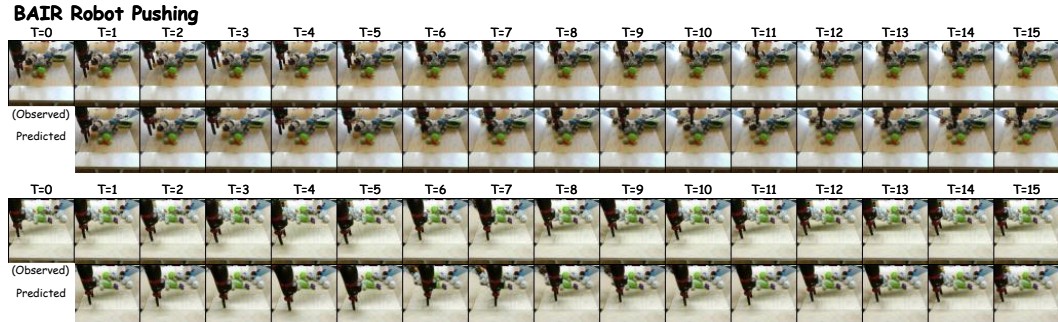

Figure 7: **Additional visualization on the BAI RRobot Pushing dataset** for action-conditioned video generation in low resolution (64 × 64).

---

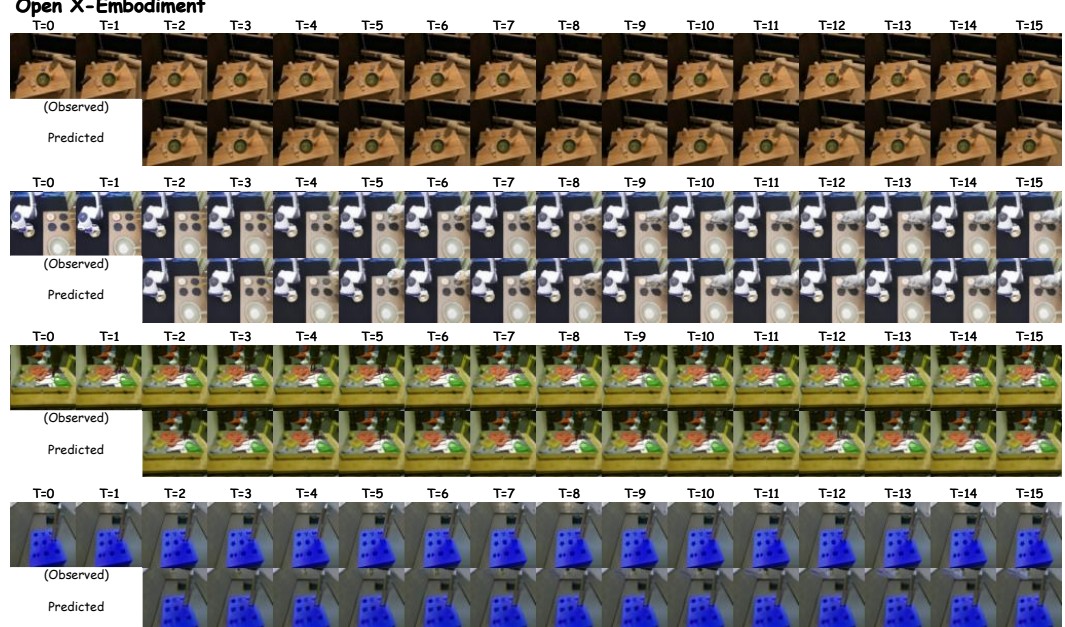

[2]Dataset and code are available at: `https://huggingface.co/datasets/1x-technologies/world_model_raw_data` under cc-by-nc-sa-4.0 license and `https://github.com/1x-technologies/1xgpt` under Apache-2.0 license

[3]For further details, see: `https://www.1x.tech/discover/1x-world-model`

[4]Code is available at: `https://github.com/s-tian/vp2`

Figure 8: **Additional visualization on the Open X-Embodiment dataset** for action-free pretraining in low resolution (64 × 64).

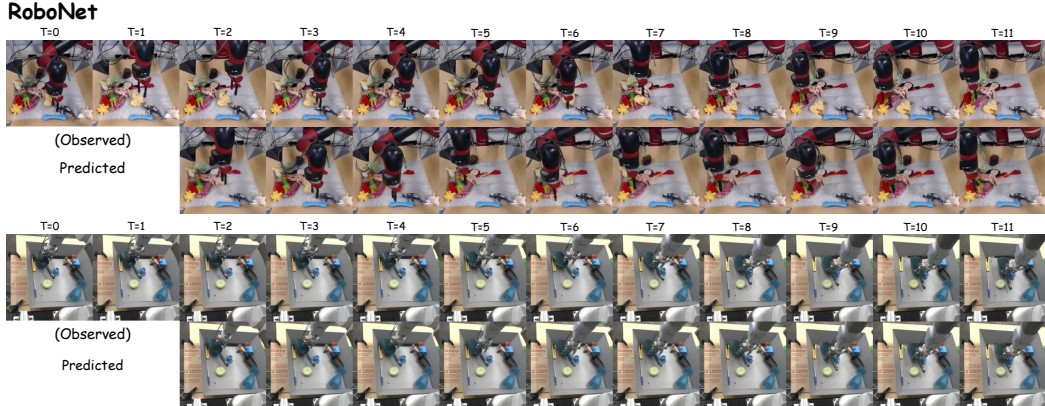

Figure 9: **Additional visualization on the RoboNet dataset** for action-conditioned video generation in high resolution (256 × 256).

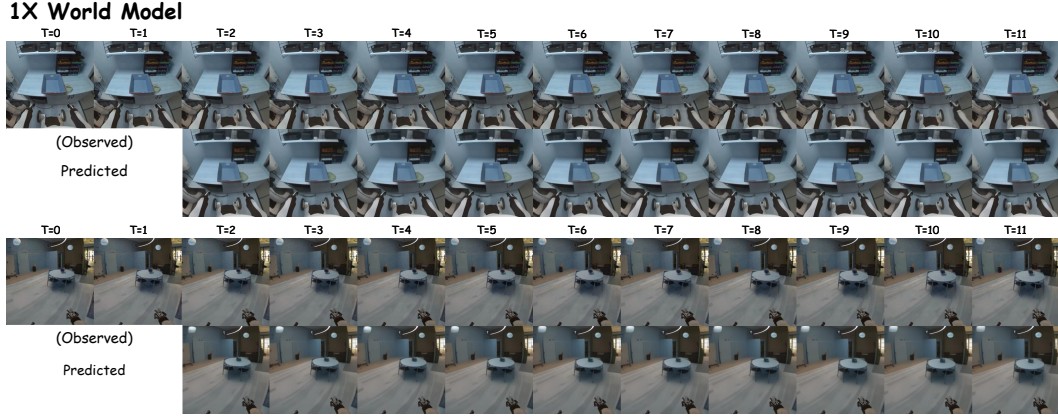

Figure 10: **Additional visualization on the 1X world model dataset** for action-conditioned video generation in high resolution (256 × 256).

# VP² RoboDesk

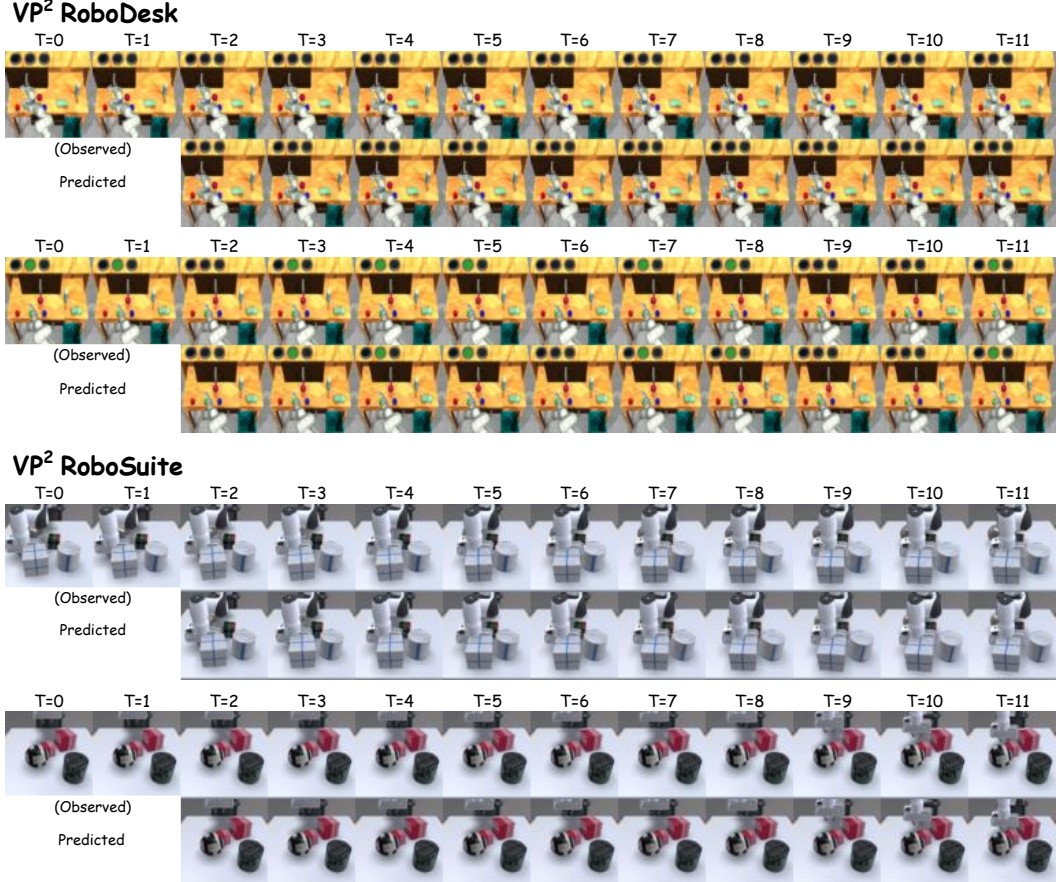

Figure 11: **Additional visualization on the VP² benchmark** for action-conditioned video generation in low resolution (64 × 64).

# Meta World

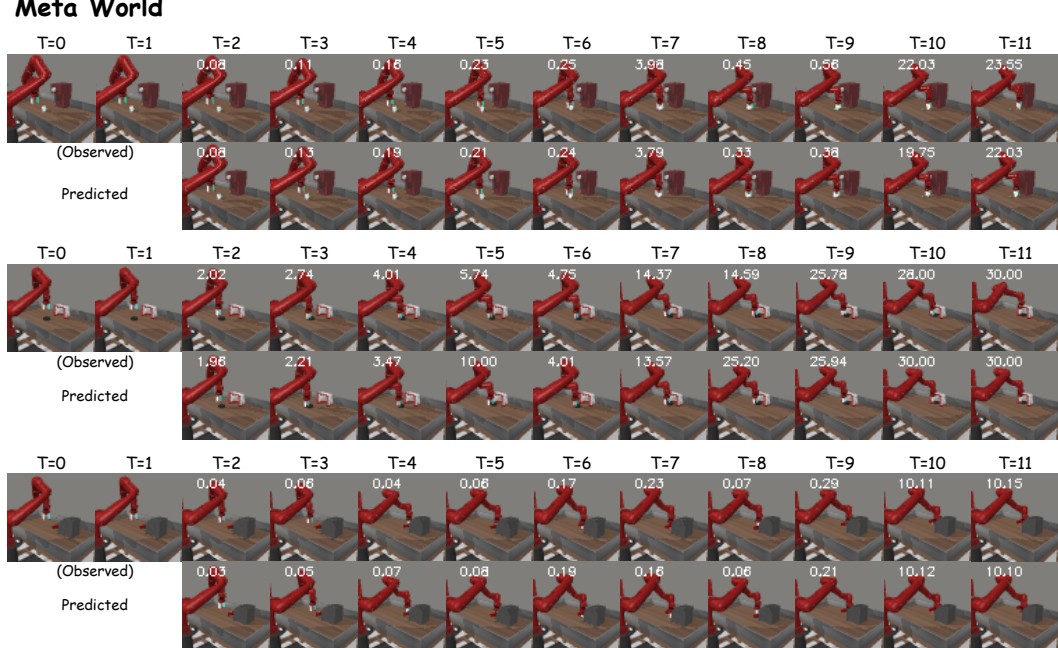

Figure 12: **Additional visualization on Meta-World tasks** for action-conditioned video generation at low resolution. Each row corresponds to a different task: *Coffee Push*, *Plate Slide*, *Handle Pull Side*. Both actual reward and predicted reward are shown in the upper left corner.

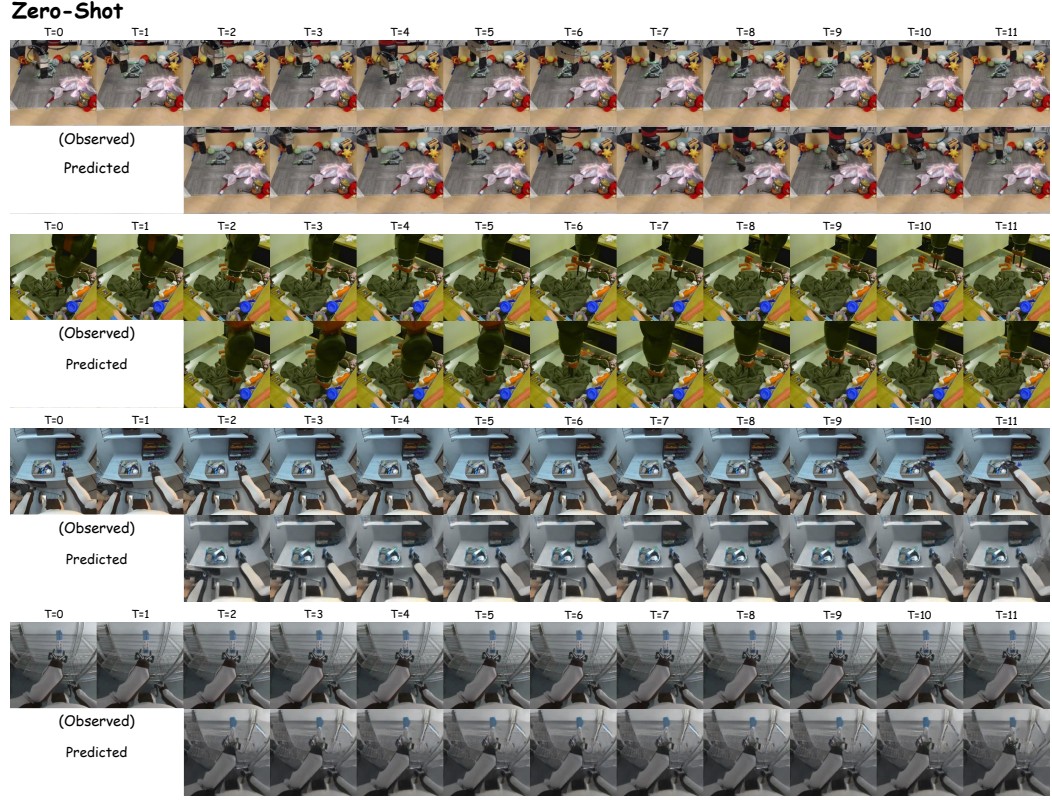

Figure 13: **Additional visualization of zero-shot performance** for action-free video generation in high resolution. The first two rows show zero-shot results on the RoboNet dataset, while the last two rows illustrate performance on the 1X dataset. Pretraining was conducted on the Open X-Embodiment dataset. The results in this figure supplement Fig. 6 in the main paper.

Motion prompts

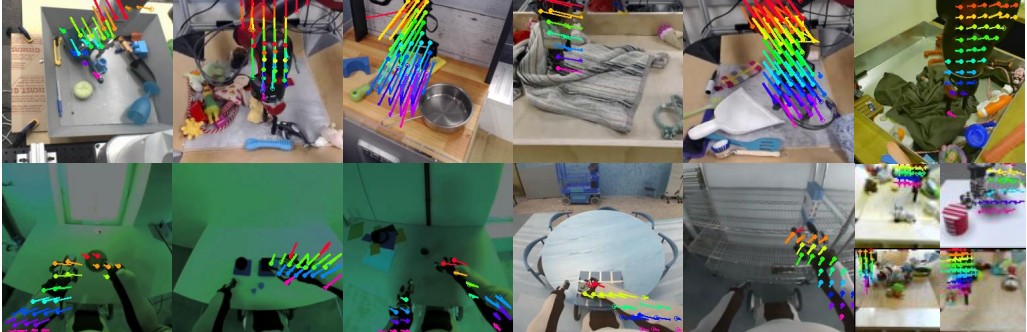

Figure 14: **Additional visualization of motion prompts** at varying resolution. Motion prompt examples are visualized on the RoboNet, 1X, BAIR, and $VP^2$ datasets. The image in the bottom right corner is shown at a resolution of $64 \times 64$. For high-resolution images ($256 \times 256$), the grid size is set to 16, while for low-resolution images ($64 \times 64$), the grid size is set to 12, supplementing the content of Sec. 3.3 in the main paper. Zoom in for a better view

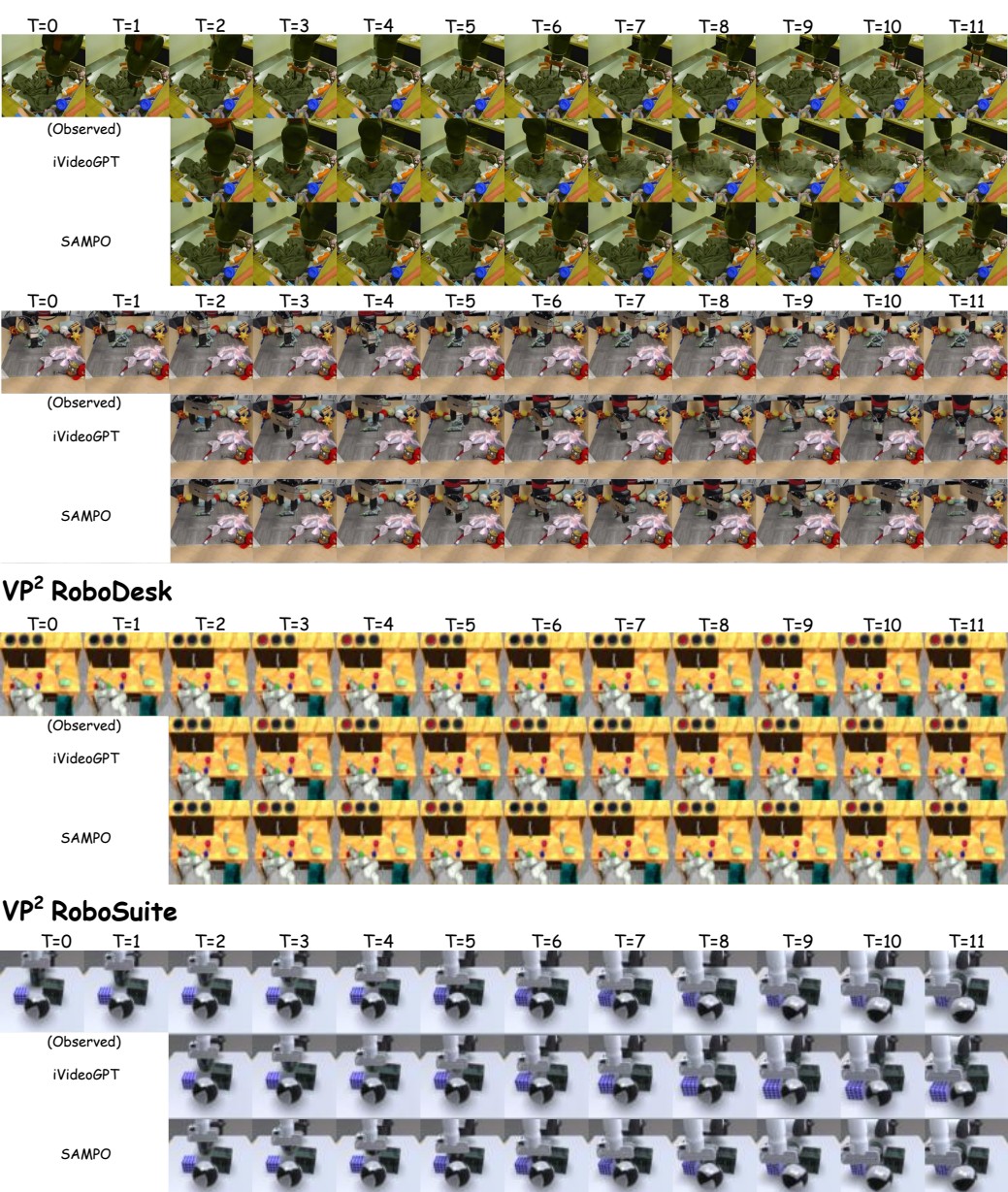

Figure 15: **Additional visual comparison with iVideoGPT** for action-conditioned video generation on RoboNet (256 × 256) and VP2 (64 × 64) datasets. The first column shows the ground truth (GT), the second column displays the predictions made by iVideoGPT, and the final column presents our predictions. As shown, our results not only effectively maintain spatial coherence in visually cluttered environments, but also better align with the ground truth motion trajectories, indicating better performance in capturing scene dynamic behaviors. This figure supplements Fig. 4 in the main paper, providing further evidence of our model's enhanced accuracy in trajectory prediction.

# D   Limitations and Future Work

**Single-frame Input Limitation.** The current motion prompt method relies on multi-frame observation sequences to extract motion trajectories (*e.g.*, point tracking for $t = 1$ to $T$ frames using CoTracker3 as described in Sec. 3.3). However, when the input consists of only a single frame ($T_0 = 1$), it is impossible to generate effective motion prompts, as trajectory extraction requires at least two frames to compute the displacement.

This limitation essentially stems from a fundamental issue in the setting: for a world model, a single frame only provides a 'starting point' for a static scene, whereas the diversity of dynamic interactions (*e.g.*, a robotic arm that may move in different directions) results in a multimodal distribution of future states. Consequently, the model needs to rely on random sampling or implicit prior generation of possible results.

To address this, potential improvements could involve the design of an implicit dynamic prior based on a single frame (*e.g.*, generating pseudo-trajectories via geometric constraints or a physics engine), or introducing a stochasticity control mechanism that balances generation diversity with physical plausibility.

**Motion Prompt as a Way of Incorporating Historical Information.** In Sec. 3.1 of the paper, the world model is formalized as a partially observable Markov decision process (POMDP), where the integration of historical information plays a crucial role in enhancing the model's ability to predict future states and develop more meaningful strategies.

In Sec.3.3, the trajectory-aware motion prompt is introduced as an external module designed to inject historical information into the hybrid autoregressive framework. While this approach has been shown to improve performance through ablation studies (Tab.4), it currently serves as an intuitive, but not necessarily the most optimal, solution for incorporating historical data.

Future work could explore more refined alternatives, such as:

1. **Implicit Dynamic Modeling:** Directly learning spatiotemporal saliency via attention mechanisms without explicit trajectory extraction (*e.g.*, combining neural differential equations to model continuous motion fields);
2. **End-to-End Motion Guidance:** Integrating trajectory prediction heads into the backbone network for joint optimization of both motion prompts and frames generation;

# E   Broader Impact

**Broader Impact of SAMPO Model.** The research presented in this paper introduces SAMPO, a model that enhances both the quality of generated content and the speed of inference, offering significant advantages over traditional autoregressive models. By mitigating issues such as object disappearance and inaccurate predictions commonly encountered in previous models, SAMPO ensures higher-quality generation. Moreover, its fast inference capabilities enable real-time decision-making and dynamic environment interactions, making it a promising solution for applications in robotic control, video prediction, and model-based reinforcement learning.

**Potential Negative Societal Impacts.** However, as with any powerful generative model, there are potential negative societal impacts. One concern is the misuse of the technology in creating deepfakes or generating realistic video content for disinformation, surveillance, or manipulation purposes. While SAMPO's primary application is in improving robotic systems, its underlying techniques could be applied in harmful ways if left unchecked.

**Mitigation Strategies for Responsible Use.** To mitigate these risks, it is essential to consider safeguards such as restricted access to the model, robust detection systems for misuse, and ethical guidelines for deployment. Additionally, ensuring transparency in the development and application of such models and creating oversight mechanisms will help prevent unintended societal consequences. Future work should also explore the potential for bias in the model's predictions, ensuring fairness and ethical deployment across diverse groups and settings.

