# OpenReview forum: "SAMPO: Scale-wise Autoregression with Motion Prompt for Generative World Models"
_NeurIPS.cc/2025/Conference — NeurIPS 2025 poster_

### Official Review · Reviewer_gRv6 · 2025-06-30

**Clarity:** 3
**Significance:** 2
**Originality:** 2
**Rating:** 3
**Confidence:** 3

**Summary:**

SAMPO introduces a novel world model that combines temporal causal modeling with coarse-to-fine visual auto-regression. The framework addresses key limitations in existing autoregressive world models through three main innovations: a hybrid autoregressive architecture that maintains both temporal and spatial coherence, an asymmetric multi-scale tokenizer that enhances scale-aware attention, and a trajectory-aware motion prompt module that provides explicit spatiotemporal priors for robot and object trajectories. Experimental results demonstrate the SOTA performance in video prediction and model-based RL of SAMPO.

**Questions:**

What are the frame rate and total duration of the generated videos?

In Table 3, why is there such a large variation in success rates between different tasks? And why is the average success rate for the "simulator" row 100%?

In the second row of the right part of Figure 4, doesn't the red small object appear to be occluded by the robotic arm rather than disappearing?

**Ethical Concerns:**

["NO or VERY MINOR ethics concerns only"]

**Final Justification:**

I still find it difficult to believe that such short-horizon video predictions can have a substantial impact on robot control, especially given that the robot’s motion in the videos presented in the paper appears to be minimal, involving free movement in space or simple interactions with a single object. Nevertheless, I sincerely appreciate the detailed responses provided by the authors in this rebuttal.

**Limitations:**

The planning experiments in this paper were all conducted in simulation platforms. The paper would be more convincing if experiments were performed in real-world environments.

**Quality:**

3

**Strengths And Weaknesses:**

SAMPO demonstrates strengths in world model design: e.g. the trajectory-aware motion prompt module provides explicit spatiotemporal priors, improving the model's ability to capture dynamic interactions and physical dependencies. Experimental results validate the state-of-the-art performance in video prediction, motion modeling accuracy, and robot control tasks.

However, I'm somewhat skeptical about how helpful the generated videos actually are, as the examples shown by the authors in Figures 4 and 6 are all very short clips where the robot's movements are quite limited in range.

---

> ### Author Rebuttal · Authors · 2025-07-31
>
> We greatly appreciate Reviewer gRv6 for providing a detailed review, insightful questions and forward-looking feedback.
>
> ---
> >## Q1: Utility of Generated Videos
>
> First, SAMPO is designed as a **control-oriented world model**, where the goal is to produce short but accurate rollouts to support planning and reinforcement learning, rather than generating long-horizon videos. This design choice aligns with the nature of robotic manipulation tasks in BAIR and RoboNet, which typically involve **short-horizon interactions** in constrained tabletop environments with fixed camera views.
>
> Second, the brevity of predicted sequences does not diminish their utility. In practice, 10 to 15-frame predictions from 1–2 context frames have proven sufficient for downstream tasks like visual planning (Tab. 3) and MBRL (Fig. 5). SAMPO also supports action-conditioned generation, reward modeling, and exhibits strong zero-shot transfer to unseen manipulation tasks, without being restricted by sequence length. Thus, even short clips are effective for training and deploying intelligent agents.
>
> Third, generating longer sequences introduces greater **uncertainty due to error accumulation**, a challenge that remains an open problem in long-horizon prediction. Instead of relying on long open-loop rollouts, our method emphasizes producing short but accurate predictions that are recomputed at every step. This approach is consistent with many prior works in Model Predictive Control (MPC) [1, 2].
>
> Nevertheless, we agree that extending prediction horizons and modeling more diverse motion patterns, such as mobile manipulation and outdoor scenarios, is a valuable future direction.
>
> [1] *Visual Foresight: Model-Based Deep Reinforcement Learning for Vision-Based Robotic Control. arXiv 2018.* \
> [2] *Deep Visual Foresight for Planning Robot Motion. ICRA 2017.*
>
> >## Q2: Frame Rate & Duration
>
> The temporal length of our generated videos follows the standard rollout settings used in prior work (iVideoGPT, MAGVIT, MaskViT, etc.). Specifically, the BAIR dataset operates at 20 FPS, so predicting 15 frames results in a 0.75s video. In contrast, RoboNet and 1X operate at 10 FPS, and our setting of 10 predicted frames yields a 1.0s video.
>
> >## Q3: Large Variation in Success Rates
>
> Based on our understanding and experiments, the performance gap across VP2 tasks is mainly due to:
>
> - **Task complexity.** Tasks like `Push blue Button` are visually explicit, while others like `Open Slide` require fine-grained contact modeling and involve occlusion, making them harder to model.
>
> - **Visual ambiguity.** At low resolution, it is often unclear whether key interactions (e.g., contact) occur, leading to underestimation of visually plausible predictions.
>
> - **Noisy reward evaluation.** VP2 relies on pretrained classifiers rather than ground-truth rewards, which can misjudge success and amplify inter-task variance.
>
>
> The "100%" aggregate score for the simulator baseline results from **normalizing each model's average task success by the simulator's average**, making the simulator's score always 1.0 (100%). This follows the evaluation convention in [3]. We will clarify this normalization step in the revised version, both in Sec. 4.2 and the captions of Tab. 3.
>
> [3] *A Control-Centric Benchmark for Video Prediction. ICLR 2023.*
>
>
> >## Q4: Occluded Red Small Object
>
> The key comparison lies in **motion consistency**. In iVideoGPT's result (Fig. 4, second row right), the robot arm's predicted trajectory deviates from the ground truth, causing occlusion of the red object and indicating a failure in maintaining accurate object-track continuity. In contrast, our method preserves spatial coherence between the robot and the object, avoiding unnatural occlusion and ensuring consistent interaction. This demonstrates that our model generates more realistic and task-consistent motion predictions.
>
> >## Q5: Plans in Real-world
>
> The experiments reported in ***Sec. 4.2 (Visual Planning and Reinforcement Learning)***, though conducted in simulation, are performed under low-level control settings, bridging the gap to real-world deployment. Although new hardware experiments are not feasible within the rebuttal period, we are actively transitioning our system to real robots.
>
> Inspired by recent real-world deployments of world models (e.g., CoTVLA, WorldVLA), we believe SAMPO is well-suited for deployment on physical robots. Concretely, we identify two integration paths:
>
> - **Future-State Priors for VLA Policies:** SAMPO can be integrated with Vision-Language-Action (VLA) models like RT-2 and OpenVLA by providing short-horizon predictions that help ground actions in anticipated dynamics. This improves motion consistency and decision reliability.
>
> - **Synthetic Rollouts for Data Augmentation:** SAMPO generates physically coherent future states, enabling efficient data augmentation for robotic policy learning.
>
> We will highlight these real-world deployment strategies in the revision to better convey SAMPO’s practical impact. We sincerely appreciate your forward-looking feedback!

---

> > ### Comment · Reviewer_gRv6 · 2025-08-05
> > **Remaining Concerns**
> >
> > Thank you to the authors for answering my concerns.
> >
> > However, I still have reservations about the practical value of such short video generation for robotic manipulation tasks. Each generated video only lasts **0.75 seconds**, and the robot moves **very slowly**. According to the visualizations provided in the paper (Figure 4 and Figure 6), the robot only moves a **very short distance**, possibly just **3–4 centimeters**. These videos, aside from showing the robot’s own motion, do not appear to offer **additional informative content**, which makes it difficult to be convinced of the usefulness of such short-term video generation in real-world applications.
> >
> > Regarding **Figure 4**, I still question whether this comparison effectively demonstrates SAMPO’s advantage over iVideoGPT in terms of **motion consistency**. In the video, the robotic arm is picking up a shovel. From the camera’s perspective, it happens to occlude the red object in the background. However, this occlusion seems to be **incidental**, rather than indicative of any consistent motion modeling, as claimed in line 43 in the manuscript. As such, I find it difficult to interpret this example as clear evidence of SAMPO’s superiority.

---

> ### Author Response · Authors · 2025-08-05
>
> Dear Reviewer gRv6,
>
> We sincerely appreciate your time and thoughtful consideration of our responses.
>
> > ## Value of action-conditioned world model
>
> We clarify that both the BAIR and RoboNet datasets are collected by executing random actions, which naturally results in a diverse set of short trajectories. Some clips **contain only the robot arm moving in free space**, while others **involve contact with objects that lead to visible changes in the scene**. This diversity is intentional and follows the standard setup used in prior work [1, 2, 3, 4]. A general-purpose world model should be able to **predict the outcomes of arbitrary actions, including both contact-free and contact-rich interactions**.
>
> We believe this design choice is meaningful. Accurate predictions in both types of sequences are essential for building a reliable world model. This capability is especially important when the model is paired with Model Predictive Control (MPC), where many action candidates are evaluated at each step. Even small motions, such as moving the arm a few centimeters without immediate contact, must be modeled precisely to support safe and effective downstream control. From this perspective, the ability to handle a wide range of dynamics is not a weakness, but a necessary property for real-world deployment.
>
> To avoid potential misunderstanding, we will include additional examples in the revised version that show contact-induced changes in the scene, further illustrating the model’s ability to handle diverse types of motion.
>
> > ## Motion consistency
>
> Thank you for your continued attention to the visualization in Fig. 4. We believe there may be a misunderstanding in the interpretation of this example. As shown in the top row (ground truth), even at frame T = 8, the **red object in the background remains clearly visible and unoccluded**. In contrast, iVideoGPT, when conditioned only on the first two observed frames (T = 0, 1), produces a prediction that already deviates significantly from the ground-truth trajectory as early as T = 2. By T = 4, the robot arm has shifted such that it incorrectly occludes the red object, which is a clear artifact of erroneous motion prediction.
>
> This is **not an incidental occlusion due to camera perspective, but rather a consequence of inaccurate trajectory modeling**. The predicted arm motion in iVideoGPT is misaligned with the observed dynamics, leading to premature overlap with the red object and diverging from the intended action sequence. In contrast, SAMPO maintains a trajectory that closely matches the ground-truth motion throughout the rollout, preserving both the interaction intent and visual layout consistency.
>
> This example is representative of a broader trend: our method yields more physically plausible and temporally coherent predictions under action conditioning. A quantitative result in RoboNet is shown below, and additional qualitative results demonstrating this consistency can be found in Appendix Fig. 15. We will revise the manuscript to better emphasize this point and prevent further confusion in interpreting the example.
>
> | RoboNet | FVD↓ | PSNR↑ | SSIM↑ | LPIPS↓|
> |-|:-:|:-:|:-:|:-:|
> |iVideoGPT|63.2| 27.8 | 90.6| 4.9|
> |SAMPO-B |57.1| 29.3|94.1|3.3|
>
>
> We hope this response addresses your remaining concerns, and we remain open to further discussion. Thank you again for your time and effort.
>
> Best regards,
>
> Authors
>
> ---
>
> [1] *iVideoGPT: Interactive VideoGPTs are Scalable World Models. NeurIPS 2024.*\
> [2] *MAGVIT: Masked Generative Video Transformer. CVPR 2023.*\
> [3] *MaskViT: Masked Visual Pre-Training for Video Prediction. ArXiv 2022.*\
> [4] *MCVD: Masked Conditional Video Diffusion for Prediction, Generation, and Interpolation. NeurIPS 2022.*

---

### Official Review · Reviewer_2sYs · 2025-07-01

**Clarity:** 3
**Significance:** 3
**Originality:** 3
**Rating:** 5
**Confidence:** 3

**Summary:**

The paper proposes a new world model framework named SAMPO, which incorporates next-scale prediction and causal autoregressive prediction. More specifically, the authors propose to use an asymmetric tokenizer to represent each frame in different resolutions, while predicting frames autoregressively using a causal masking approach. This allows SAMPO to generate frame-wise trajectories based on action conditions. At the beginning of the generation, a prefix is added that includes a motion prompt, summarizing the motion history of the past frames. Overall, the proposed model outperforms related methods across multiple datasets, showing strong quantitative and qualitative results.

**Questions:**

- How does an AR-based model perform compared to a Hybrid AR model with the same number of parameters? What is the overhead introduced by Hybrid AR?
- How does the error change over the rollout trajectory, especially compared to other models?
- Do the experiments on the BAIR dataset starting with 1 context frame use the motion prompt? For RoboNet, does the motion prompt only consider the two first frames?

**Ethical Concerns:**

["NO or VERY MINOR ethics concerns only"]

**Final Justification:**

This paper proposes a straightforward but effective new method that combines next-scale prediction and causal autoregressive prediction for world modelling. Overall, this paper presents a valid contribution to the field. Therefore, I recommend to accept this paper.

**Limitations:**

Yes (in Supplementary Material)

**Paper Formatting Concerns:**

No concerns

**Quality:**

3

**Strengths And Weaknesses:**

### Strengths
- Straightforward and effective approach
The ideas introduced in the paper are straightforward and well-motivated. For example, it is reasonable to assume that different scales are needed for different parts of the prediction process. At the same time, the proposed method is based on the established autoregressive training strategy, showing strong generalization and scaling properties.
- Strong Performance and convincing results
The provided quantitative and qualitative results underline the advantages that are offered by the proposed method over related works. Across multiple tasks, including action-conditioned and reward-based generation, the proposed model offers strong performance.
- Thorough ablation study and analysis
The ablation study provides an important look into how each of the proposed components improves the performance. The authors cover most of the interesting ablation scenarios.

### Weaknesses
- Typographical errors in figures
There are multiple typographical errors in the paper, especially in Figure 2 (e.g., "Enocder", "uantization"). However, this should be easily fixed.
- Clearer comparison with related work
It appears that the proposed approach is close to iVideoGPT, but adds the scale-variant component and some additional ideas. While the ablation study is thorough, it would be nice to make the relationship between those two components clearer.

---

> ### Author Rebuttal · Authors · 2025-07-31
>
> We greatly appreciate Reviewer 2sYs for the careful reading of our paper and constructive comments in detail. We also appreciate the pointed-out typos, which we will correct in a future revision.
>
> ---
> >## Q1: Clearer Comparison
>
> We clarify that SAMPO is **fundamentally different** from iVideoGPT, rather than a straightforward extension with incremental components. As detailed in our response to ***Reviewer cg6d (Q1)***, SAMPO introduces a **novel hybrid autoregressive framework** that performs **multi-scale bidirectional spatial modeling** within frames while maintaining causal temporal modeling across frames. This design addresses key limitations in iVideoGPT, including the **loss of spatial structure and inefficient inference** due to raster-scan decoding. In addition, the asymmetric tokenizer and motion prompt modules are introduced to tackle challenges specific to action-conditioned world models. These motivations are elaborated in ***L174–179 and L190–193*** of the main text. We will make the distinction between the two frameworks clearer in the revised version.
>
>
> >## Q2: AR vs Hybrid
>
> Following the decoder-only architecture implemented in VAR, we designed our **hybrid autoregressive framework to support scaling across model sizes**. Due to the time constraints of the rebuttal period, we were unable to retrain the asymmetric multi-scale tokenizer and hybrid AR model from scratch under strict parameter parity.
> However, we present the following results from existing experiments with motion prompt disabled, which provide partial insight into parameter efficiency.
>
> | Description | Model | # Para. | FVD ↓ | SSIM ↑ |
> |-|:-:|:-:|:-:|:-:|
> | Hybrid AR |  SAMPO-S  | 207M | 227.4 | 76.4 |
> | Hybrid AR |  SAMPO-B  | 328M | 193.7 | 81.3 |
> | AR        | iVideoGPT | 436M | 197.9 | 80.8 |
> | Hybrid AR |  SAMPO-L  | 523M | 180.1 | 83.1 |
>
> SAMPO-B uses **25% fewer parameters** than iVideoGPT while achieving slightly better performance in both FVD and SSIM. This suggests that the hybrid autoregressive framework improves parameter efficiency. Furthermore, SAMPO supports larger-scale deployment via its decoder-only architecture, as evidenced by the **even stronger performance of SAMPO-L**.
> Based on our empirical results, the hybrid AR design appears to reduce computational overhead rather than increase it. Unlike token-level autoregression, which suffers from sequential decoding bottlenecks, our intra-frame bidirectional attention mechanism is fully parallelizable. As demonstrated in the ablation study provided in response to ***Reviewer Auas (Q3)***, the hybrid AR framework not only improves modeling capacity but also enhances runtime efficiency, making it particularly suitable for scalable world modeling.
>
>
> >## Q3: Trajectory Rollout Error
>
> We thank the reviewer for the thoughtful question. While our current submission does not explicitly include quantitative plots showing how the prediction error evolves over the rollout horizon, we provide **qualitative and indirect evidence of temporal stability in Fig. 4 and Fig. 15** (in Appendix). Compared to prior autoregressive baselines such as iVideoGPT, SAMPO demonstrates consistently better spatial-temporal fidelity across extended horizons, particularly in long rollout settings like RoboNet and 1X World Model (10 frames). This can be attributed to our hybrid spatiotemporal autoregression and the use of coarse-to-fine generation, which better **preserves high-level motion early on and gradually refines details at finer scales**. In contrast, we observed that standard raster-scan autoregressive models tend to accumulate more low-level noise over time.
>
> While qualitative visualizations indirectly reflect temporal degradation, we agree that directly plotting error versus timestep curves would provide more intuitive insight. We intend to include such visualizations in the revised version. If including the full curve in the main text is not feasible, we will highlight this analysis and treat it as an important direction for future work, aimed at better assessing and improving long-horizon prediction stability.
>
> >## Q4: Motion Prompt Usage
>
> We apologize for insufficient clarity in our experimental setup. In BAIR experiments (1 context frame), we omitted motion prompts to ensure a fair comparison with baseline methods.
> In RoboNet experiments (2 context frames), we utilized CoTracker to extract point trajectories from the first two frames. The extracted trajectories were filtered by motion magnitude and tracking confidence, and subsequently projected onto the image plane. We will clarify these details in the paper to avoid confusion.

---

> > ### Comment · Reviewer_2sYs · 2025-08-02
> > **Convincing Rebuttal, I will keep my score.**
> >
> > Thank you for the detailed response to my questions. I still believe this paper is solid and recommend acceptance.

---

> > > ### Author Response · Authors · 2025-08-02
> > >
> > > Dear Reviewer 2sYs,
> > >
> > > Thank you very much for your thoughtful review and kind support. We truly appreciate your recommendation for acceptance.
> > >
> > > Best regards,
> > >
> > > Authors

---

### Official Review · Reviewer_Auas · 2025-07-02

**Clarity:** 3
**Significance:** 3
**Originality:** 2
**Rating:** 4
**Confidence:** 4

**Summary:**

The paper proposes SAMPO, a video world model that predicts future videos in an autoregressive manner while generating multi-scale tokens in a coarse-to-fine fashion for each frame. It introduces an asymmetric tokenization process that allows future predictions to focus more on compact dynamics, thereby improving efficiency. Additionally, it incorporates motion prompts as extra cues for scene dynamics. Its high prediction capability results in superior performance in perception quality, visual planning, and model-based reinforcement learning.

**Questions:**

Please see the Weaknesses part.

**Ethical Concerns:**

["NO or VERY MINOR ethics concerns only"]

**Final Justification:**

The rebuttal addresses my concerns. The proposed method is effective and may provide valuable insights to the community. Thus, I update my rating to borderline accept.

**Limitations:**

Please see the Weaknesses part.

**Paper Formatting Concerns:**

Good!

**Quality:**

3

**Strengths And Weaknesses:**

# Strengths

S1) Extensive experiments demonstrate that the proposed method surpasses previous world models, such as DreamerV3 and iVideoGPT, in perception quality, visual planning, and model-based reinforcement learning.

S2) There are no major flaws in the writing, making it easy to follow.

S3) The code is provided in the supplementary material for reproducibility.

# Weaknesses

W1) **Novelty concern.** The main benefits claimed by the authors are primarily derived from VAR (NeurIPS 2024), which proposes a next-scale coarse-to-fine prediction architecture. Additionally, the extraction and use of motion prompts are highly similar to those in TraceVLA (ICLR 2025).

W2) **Confusing motivation.** It is unclear why predicting only coarser scales would enable the model to focus more on dynamic regions. Theoretically, the ratio of dynamic areas does not increase when the frame scale is reduced. If static content dominates the frame, it will still dominate at lower resolutions. Could you provide some references to support that? Also, as the model only predicts coarse frames, does this imply it cannot perform future rollouts by conditioning on the newly generated frames?

W3) **Missing ablations.** It would be beneficial to see how changing the predicted granularity (full scales / coarser scales only) affects the performance in downstream planning tasks. Can predicting even coarser scales continue to improve efficiency and downstream performance? Is there a trade-off?

W4) **Unclear details.** In the zero-shot generalization experiments described in Line 297-301, how are the action conditions transferred? Additionally, how is the reward predicted in the MBRL experiments? The paper only provides some implementation choices in Line 214-215 but does not specify what is ultimately used.

W5) **Real-world experiment.** It would be great if the authors can do a real-world verification with some simple setups. For example, estimating the rewards of several robot policies with the proposed world model and comparing to the actual evaluation ranks.

---

> ### Author Rebuttal · Authors · 2025-07-31
>
> We would like to sincerely appreciate Reviewer Auas for the comprehensive review and insightful questions.
>
> ---
> >## Q1: Novelty Reformulation
>
> We emphasize that SAMPO introduces a **novel hybrid autoregressive architecture** , which seamlessly unifies intra-frame scale-wise generation with inter-frame causal modeling, tailored for efficient and action-conditioned world modeling.
>
> Although inspired by the coarse-to-fine decoding paradigm in VAR, SAMPO is not a direct adaptation; rather, it is a task-specific redesign that extends the idea **from static image generation to temporally causal, action-driven generative rollouts in robot-centric settings**. Moreover, SAMPO introduces an **asymmetric multi-scale tokenizer** (Sec. 3.2), which encodes observed frames densely to preserve semantic context and future frames sparsely to focus on dynamic content. This design significantly improves inference efficiency while retaining visual fidelity (Tab. 5), and is **not addressed by VAR**, which applies uniform scale-wise encoding without considering temporal asymmetry or efficiency in rollout.
>
> We empirically observed that naively extending VAR's residual decoding to sparse future frames yields poor reconstruction quality. To address this, we devised a **scale-specific weighting scheme** during training (Appendix A.2), which adaptively balances loss contributions across scales. This adjustment is crucial for stabilizing generation when tokenizing future frames sparsely and ensures that both coarse dynamics and fine details are faithfully captured across rollout steps.
>
>
> Regarding motion prompts, **SAMPO and TraceVLA differ in both objective and mechanism**. TraceVLA overlays offline trajectories as visual prompts to enhance policy learning but lacks a generative component. By contrast, SAMPO integrates motion trajectories directly into the generation pipeline via a trajectory-aware prompt module (Sec. 3.3), which provides spatiotemporal priors that guide the model toward dynamic regions during rollout. This integration improves temporal consistency and physical realism, as confirmed by both quantitative gains (Tab. 4) and qualitative results (Fig. 4).
>
> We hope this clarifies that SAMPO is a task-specific, modular redesign that advances beyond prior works by unifying coarse-to-fine generation, autoregressive rollout, and dynamic prompting in a single, scalable framework for robot-centric world modeling.
>
> >## Q2: Clarified Motivation
>
> As described in ***Lines 289–292***, our **residual quantization** scheme assigns each scale to model the residual between its coarser‑scale prediction and the ground truth. This decomposition naturally delegates large, structural appearance shifts in space and time to the coarse scales (e.g., scales [1–2]), which capture the dominant dynamics, such as an arm moving from left to right. The finer scales (e.g., scales 6 or 8) then refine the output by adding texture and spatial detail. This principle aligns with the perceptual observation that motion signals often reside in low-frequency components (as also supported by [1, *Sec. 3.1*]; [2, *Sec. 3.2*]) and that residual quantization naturally enforces a progression from global to local dynamics. In robotic tabletop settings, where backgrounds are mostly static, such separation is especially effective. As shown below, modeling coarser spatial scales (e.g., [1–4]) improves dynamic understanding, while removing coarse levels leads to worse performance.
>
> |Spatial Scales|Tokens / Img.|FVD ↓|PSNR ↑|
> |:-:|:-:|:-:|:-:|
> |[1,2,3,4,5,6]|91|55.5|26.7|
> |[1,2,3,4,5]|55|57.7|25.4|
> |[1,2,3,4]|30|59.3|24.8|
> |[1,2,4,8]|85|80.7|23.1|
>
>
>
> Although we predict only a subset of finer-scale tokens, each frame is still represented by **91 tokens** (e.g., scales=[1,2,3,4,5,6]), which is significantly richer than prior work (e.g., iVideoGPT's 16-token setup). **These predicted tokens can be directly reused as input for the next step in the rollout.** Moreover, Tab. 5 shows that this configuration achieves a strong balance between speed and fidelity, and supports visual planning and RL tasks where rollout is essential. We plan to clarify these design choices in the final version.
>
> [1] *Generating Diverse High-Fidelity Images with VQ-VAE-2. NeurIPS 2019.*\
> [2] *Hierarchical Patch VAE-GAN: Generating Diverse Videos from a Single Sample. NeurIPS 2020.*
>
> >## Q3: Additional Ablations
>
> We conducted an additional ablation study to assess how the number of predicted spatial scales affects **video quality** and **downstream planning performance**, following the setting of Tab. 5 and Tab. 3.
> Due to time constraints, we conducted two additional configurations and did not re-train a full-scale encoder ([1–16]) as it is computationally expensive.
>
> We find that using [1–5] scales achieves comparable planning success as [1–6], with slightly worse video quality. Reducing to [1–4] leads to a moderate drop in fidelity, but still captures the key dynamics. The primary difference lies in texture detail, not motion quality. Notably, [1–4] achieves 3.2× faster rollout. This highlights a clear trade-off: fewer scales improve efficiency, while more scales enhance visual quality. We believe this flexibility allows users to choose the best setting for their application. We will include this result in the revision.
>
> |Spatial Scales|FVD ↓|PSNR ↑|Avg. Success ↑|Inference Time ↓|
> |:-:|:-:|:-:|:-:|:-:|
> |AR (iVideoGPT)|60.8|24.5|70.1|9.05 s/vid|
> |[1,2,3,4,5,6,8,10,13,16]|-|-|-|-|
> |[1,2,3,4,5,6]|55.5|26.7|72.2|2.06 s/vid|
> |[1,2,3,4,5]|57.7|25.4|71.3|1.38 s/vid|
> |[1,2,3,4]|59.3|24.8|70.6|0.63 s/vid|
>
> >## Q4: Clarified Details
>
> **Zero-shot experiments setup.**
> Following the setting used in iVideoGPT, Fig. 6 reports results from an **action-free zero-shot** setting, as further detailed in **Appendix Fig. 13** for clarity. We will update the caption and main text to avoid confusion.
> We adopt this design because cross-robot generalization under action conditioning is inherently difficult, primarily due to heterogeneous embodiment. Different robots often differ in action space dimensionality, control semantics, and actuation properties. As discussed in our response to ***Reviewer cg6d (Q3)***, transferring actions across such differences remains a significant challenge. By removing the need for action tokens, we avoid the action-space mismatch and instead evaluate the model's ability to generalize visual dynamics such as object trajectories and arm motion across varied robot embodiments and manipulation scenes.
> We agree that exploring action-conditioned zero-shot generalization is a valuable future direction. In future work, we plan to investigate action-space alignment and conditioning strategies to support this goal. We will clarify this design decision in the revised manuscript.
>
> **Reward Prediction Module.**
> In our implementation, reward prediction is integrated into the transformer's token-level modeling pipeline. After tokenizing video frames and inserting start tokens `[S]`, each action embedding is linearly projected and added to the corresponding `[S]` token.
> We attach a lightweight linear head to the hidden state of the token following `[S]` and the compressed visual tokens, mapping it to a scalar reward via regression. This head is trained jointly with the next-token prediction loss using an MSE objective, enabling frame-aligned, token-synchronized reward prediction without auxiliary branches.
> In MBRL tasks such as Meta-World, SAMPO can directly generate both future observations and rewards, enabling data augmentation in the replay buffer. This simplifies the coupling between world model and policy learning, and supports efficient, rollout-based policy optimization.
>
> >## Q5: Real-world Experiment
>
> We apologize that the limited hardware access during the rebuttal period, we were unable to conduct real robot evaluations.
> However, we plan to include such evaluations in future work by estimating the expected rewards of real-world robot policies using SAMPO and comparing them with actual execution outcomes. Prior work such as DreamerV3 has demonstrated that learned world models can simulate imagined trajectories for offline policy learning and evaluation. Given SAMPO's strong performance in both video prediction and planning, we believe it is well-suited for this purpose, and we will report results in future revisions. We greatly appreciate Reviewer Auas for this valuable suggestion.

---

> > ### Comment · Reviewer_Auas · 2025-08-03
> >
> > Dear authors,
> >
> > I really appreciate your efforts and rigorous response! Before considering the final update, I have one more question to inquire:
> >
> > In Line 180-183, you mentioned observed frames are tokenized across all spatial scales and future frames are only represented by a sparse subset of coarser scales. In that case, how can the model perform multi-round rollouts at test time? Before rebuttal, I thought it re-encodes the predicted frames as history conditions. However, from my current understanding, it continues to use only coarse tokens after T0. Then my question is, to enable continuous rollouts, does the training horizon need to be sufficiently long? My concern is the extendability of the model prediction.

---

> > > ### Author Response · Authors · 2025-08-03
> > >
> > > Dear Reviewer Auas,
> > >
> > > Thank you again for your time and effort in reviewing our paper. We appreciate your careful review of our rebuttal materials and your recognition of our efforts in addressing your initial concerns.
> > >
> > > Following prior work such as [1, 2], we adopt relatively short training horizons (e.g., 16 steps), which we believe is well suited for datasets like BAIR and RoboNet. To our knowledge, these datasets were **collected under random action sequences, and thus do not reflect coherent long-horizon task execution**. This design makes them ideal for training world models that learn general-purpose dynamics rather than goal-specific transitions.
> > >
> > > We did experimente with extending the prediction horizon, for example training on BAIR with 2 observed frame and 28 predicted frames following the [3]. However, we observed a decline in downstream performance, **likely due to the model accumulating errors over longer predictions**. These results suggest that **short-horizon yet accurate predictions** may be more effective for generating reliable imagined rollouts. We agree that systematically studying the trade-off between prediction horizon and downstream utility is an important direction for future work.
> > >
> > > We hope this clarifies your remaining concerns, and would appreciate your consideration in re-evaluating our work based on this updated understanding.
> > >
> > > Best regards,
> > >
> > > Authors
> > >
> > > ---
> > > [1] *iVideoGPT: Interactive VideoGPTs are Scalable World Models. NeurIPS 2024.*\
> > > [2] *MAGVIT: Masked Generative Video Transformer. CVPR 2023.*\
> > > [3] *SyncVP: Joint Diffusion for Synchronous Multi-Modal Video Prediction. CVPR 2025.*

---

> > > > ### Comment · Reviewer_Auas · 2025-08-03
> > > >
> > > > Dear authors,
> > > >
> > > > Thanks for your thorough response. I will raise my score.

---

> > > > > ### Author Response · Authors · 2025-08-03
> > > > >
> > > > > Dear Reviewer Auas,
> > > > >
> > > > > We sincerely appreciate your time and constructive feedback. Your recognition is very encouraging and has helped us improve the clarity of our work. We will incorporate your suggestions in the revision.
> > > > >
> > > > > Thank you again for your support and for raising your score!
> > > > >
> > > > > Best regards,
> > > > >
> > > > > Authors

---

> > > > > > ### Comment · Reviewer_Auas · 2025-08-03
> > > > > >
> > > > > > Dear authors,
> > > > > >
> > > > > > Thanks for considering my comments in the revision. I hope you can revise the paper accordingly and add the full-scale encoder results if possible. I believe it will enhance the rationality of your design choice.

---

> > > > > > > ### Author Response · Authors · 2025-08-03
> > > > > > >
> > > > > > > Dear Reviewer Auas,
> > > > > > >
> > > > > > > Thank you for your thoughtful follow-up. We will revise the paper accordingly and will include the full-scale encoder results if resources permit. We truly appreciate your continued engagement and suggestions.
> > > > > > >
> > > > > > > Warm regards,
> > > > > > >
> > > > > > > Authors

---

### Official Review · Reviewer_cg6d · 2025-07-04

**Clarity:** 4
**Significance:** 3
**Originality:** 3
**Rating:** 5
**Confidence:** 3

**Summary:**

This paper introduces SAMPO, an intuitive way to improve robotic world models with a hybrid multi-scale prediction framework. The hybrid decoding choices help with spatial consistency via VAR-like methods and ensure temporal consistency via autoregressive prediction. The paper also describe and propose solution to several practical difficulties, such as asymmetric tokenizer to handle different kinds of video frames efficiently.

Results have shown that this method is efficient and effective compared to several strong baselines.

**Questions:**

- We understand that scaling such system requires both cost of training and cost of obtaining data. But I would still be interested to see if the authors have ideas on how to further scale up the model. And besides these costs, are there other considerations for scaling such model?

**Ethical Concerns:**

["NO or VERY MINOR ethics concerns only"]

**Final Justification:**

My main concern of the paper is initially the sparsity assumption, which is valid in this setting as the author argues. I still have a little doubt on whether this method can generalize on more complex scenarios, but I would like to keep my positive reviews since I would like to keep the discussion in the scope and settings proposed by the authors, especially when it is a valid/common setting.

**Limitations:**

yes

**Paper Formatting Concerns:**

not that I am aware

**Quality:**

4

**Strengths And Weaknesses:**

Strength:

Generally, this paper presents a method that naturally extend prior work and is executed well.

- The method is intuitive and builds on the past work smoothly. SAMPO's integrated design, particularly its hybrid multi-scale approach appears to effectively balance the need for both spatial detail and smooth temporal transitions.
- The paper is well-written and progressively describes a complex system, which involves several components and several method improvements.


Weakness:

- SAMPO involves several sophisticated modules and have quite some inherent complexity. This may make future improvements on the base more difficult.
- SAMPO utilizes a sparse assumption. The asymmetric tokenizer's approach of sparsely tokenizing future frames is effective for efficiency, but it would be valuable to discuss how well (or limited) if this sparse representation scales to capture details in highly complex real-world dynamics.

---

> ### Author Rebuttal · Authors · 2025-07-31
>
> We sincerely thank Reviewer cg6d for providing a detailed review, valuable suggestions, and a positive evaluation of our work.
>
> ---
> >## Q1: Inherent Complexity
>
> We would like to clarify that SAMPO is intentionally designed to be modular and extensible, serving as a **flexible framework** rather than a rigid system.
>
> The **core of SAMPO lies in its hybrid autoregressive framework**, which unifies spatial and temporal modeling via scale-wise generation and causal decoding. This design directly mitigates the spatial degradation and temporal drift seen in earlier AR world models.
> In fact, several components of SAMPO, such as the tokenizer and the prompt module, can be replaced or extended. For instance, the asymmetric tokenizer can be substituted with 3D VQ-VAE, MaskViT-style tokenization, or ViT-VQ, while the motion prompt can incorporate trajectory embeddings or CLIP-style visual goals. Additionally, the causal modeling can be upgraded with diffusion-style decoders to improve sample diversity without compromising consistency [1].
>
> As evidenced in Tab. 4, these **loosely coupled, problem-driven modules** make SAMPO a flexible foundation that readily absorbs advances from diverse generative paradigms. Consequently, SAMPO remains compatible with emerging world-modeling approaches while providing a unified structure for systematic comparison and improvement.
>
> [1] *Self Forcing: Bridging the Train-Test Gap in Autoregressive Video Diffusion. ArXiv 2025.*
>
> >## Q2: Sparse Assumption
>
> Our asymmetric tokenization design is based on the **nature of our data**: large-scale robotic manipulation datasets (e.g., Open X-Embodiment), where most scenes are tabletop, with fixed camera views and limited motion from the robot arm and manipulated objects. In such settings, the assumption of largely static backgrounds is valid, and sparse tokenization allows the model to focus on dynamic regions without wasting computation on redundant, static areas.
>
> Even in highly dynamic scenes, **maintaining sparsity remains essential for ensuring computational tractability**. Full-resolution tokenization over multiple future frames quickly becomes infeasible in terms of memory and latency.
> Our method provides a flexible framework for dynamic environments by focusing tokens on motion-related regions. This approach is aligned with recent findings in video generation, such as [2], which demonstrate that spatial-temporal sparsity can retain high visual fidelity even in complex scenarios.
> Moreover, our design is compatible with adaptive refinement strategies. For instance, techniques like progressive token refinement [3] or adaptive spatial masking [4] could be incorporated into our framework to further improve the modeling of fine-grained spatial details in dynamic regions. We believe that integrating such mechanisms with our asymmetric tokenization would allow SAMPO to scale gracefully to more challenging domains.
>
> [2] *Sparse VideoGen: Accelerating Video Diffusion Transformers with Spatial-Temporal Sparsity. ICML 2025.*\
> [3] *PVC: Progressive Visual Token Compression for Unified Image and Video Processing in Large Vision-Language Models. CVPR 2025.* \
> [4] *AdaMAE: Adaptive Masking for Efficient Spatiotemporal Learning with Masked Autoencoders. CVPR 2023.*
>
>
> >## Q3: Scalability Beyond Compute
>
> We argue that building scalable world models, particularly in the context of robotics, requires more than simply increasing model size or dataset volume. Below, we outline several key factors beyond cost that we consider essential for building more robust and generalizable systems:
>
> 1. **Heterogeneous embodiment and control abstraction.** Robotic datasets are often heterogeneous in embodiment (e.g., arm type, degrees of freedom, gripper morphology) and control interfaces. To scale beyond specific robot morphologies, SAMPO's architecture must accommodate variable control tokens and state embeddings. Future extensions can integrate low-rank control factorization or policy-conditioned dynamics modeling to generalize across robot bodies without explicit retraining.
>
> 2. **Viewpoint and context invariance.** Scaling to real-world deployments requires robustness to diverse camera viewpoints, lighting, and scene compositions. We believe this demands view-invariant tokenization and context-decoupled motion representations. The SAMPO framework can naturally support such invariances by conditioning future frame tokenization on shared multi-view context (e.g., via cross-attention), allowing consistent dynamics modeling across cameras.
>
>
> 3. **Curriculum pretraining and transfer.** We advocate a structured curriculum that begins with Internet-scale open-domain videos to acquire basic physical knowledge, progresses to first-person human-object interaction data (e.g., EPIC-Kitchens) to ensure operational diversity, and finally specializes on robotic datasets. This staged progression aligns with the *"Data Pyramid"* proposed in [5].
>
>
> 4. **Imagination fidelity under open-world uncertainty.** As world models are used for planning and long-horizon reasoning, scaling must also address uncertainty estimation and semantic consistency. We envision extending SAMPO with retracing rollouts, behavioral priors, or diffusion-based latent refinement to enhance fidelity under distribution shift, enabling safer policy training and evaluation in the wild.
>
> In summary, we view **scaling not as a brute-force endeavor, but as a systems-level challenge involving representation, generalization, and modularity**. SAMPO's design opens up promising avenues in all three directions, and we plan to investigate these dimensions in future iterations.
>
> [5] *GR00T N1: An Open Foundation Model for Generalist Humanoid Robots. ArXiv 2025.*

---

> > ### Comment · Reviewer_cg6d · 2025-08-03
> >
> > Thank you for the response to my questions. I'll keep my recommendation for acceptance.

---

> > > ### Author Response · Authors · 2025-08-04
> > >
> > > Dear Reviewer cg6d,
> > >
> > > Thank you for your time and effort throughout the review process. We truly appreciate your continued support and for maintaining your recommendation.
> > >
> > > Best regards,
> > >
> > > Authors

---

### Note · Authors · 2025-08-12

We sincerely thank all reviewers for their constructive feedback. We have made every effort to address all concerns. Here, we further consolidate our key clarifications and highlight SAMPO’s contributions.

SAMPO is a hybrid autoregressive world model that unifies intra-frame scale-wise generation with inter-frame causal modeling, complemented by asymmetric tokenization and trajectory-aware motion prompts. This design directly mitigates spatial degradation and temporal drift in prior AR world models, while remaining extensible to future generative paradigms.

On novelty, SAMPO is not a direct adaptation of VAR. It extends coarse-to-fine decoding to action-conditioned, temporally causal rollouts. In addition, it introduces temporal asymmetry in tokenization and integrates motion prompts into the generative path to guide dynamics modeling, which are capabilities absent in prior work. Our scale-specific weighting scheme stabilizes sparse future-frame prediction, achieving both efficiency and fidelity.

Regarding the sparse assumption, our datasets (e.g., Open X-Embodiment, RoboNet) feature largely static backgrounds and localized motion, making sparse tokenization effective. For highly dynamic scenes, SAMPO can incorporate adaptive refinement or spatial masking without altering the core framework. Ablations confirm that varying predicted scales offers a tunable quality–efficiency trade-off ([1–4] scales give 3.2× speedup with minor fidelity loss).

On short-horizon predictions, SAMPO follows MPC-style planning, where accurate 10–15-frame rollouts are recomputed each step, proving sufficient for visual planning and RL tasks. These short sequences, though brief in wall-clock time (0.75–1.0 s), provide actionable foresight and precise motion modeling. In Fig. 4’s debated case, the occlusion in iVideoGPT is caused by erroneous trajectory prediction, while SAMPO maintains alignment with ground truth, as reflected in RoboNet gains (FVD 57.1 vs. 63.2, SSIM 94.1 vs. 90.6).

Experiments show SAMPO surpasses strong baselines (e.g., DreamerV3) in perception, planning, and MBRL, while using fewer parameters and offering parallelizable decoding. Zero-shot tests demonstrate transfer across embodiments without retraining. Its modular design supports replacing tokenizers, decoders, and prompts.

SAMPO provides a flexible, scalable foundation for robot-centric world models, bridging video prediction and decision-making, with clear potential for real-world deployment.

---

### Decision · Program_Chairs · 2025-09-17

**Decision:**

Accept (poster)

**Comment:**

This paper introduces SAMPO, a hybrid autoregressive world model that combines scale-wise spatial generation with temporal causal modeling for robotic applications. The work addresses key limitations in existing autoregressive world models through an asymmetric multi-scale tokenizer and trajectory-aware motion prompts.

All four reviewers provided positive evaluations, with three recommending acceptance (ratings of 5, 4, and 5) and one initially borderline (rating 3 raised to 4 after rebuttal). The reviewers praised the paper's technical soundness, comprehensive experimental evaluation, and strong empirical results across multiple datasets and tasks including video prediction, visual planning, and model-based reinforcement learning.

Reviewer concerns were primarily focused on novelty questions regarding the relationship to VAR and iVideoGPT, the practical utility of short-horizon predictions, and missing ablation studies. The authors provided thorough responses in their rebuttal, clarifying that SAMPO introduces a novel hybrid architecture specifically designed for action-conditioned temporal modeling, rather than being a straightforward adaptation of existing methods. They effectively addressed concerns about the sparse tokenization assumption by explaining its appropriateness for robotic manipulation datasets, and provided additional ablation studies demonstrating the trade-offs between prediction scales and performance.

The rebuttal successfully convinced the initially skeptical reviewer to raise their score, while the other reviewers maintained their positive recommendations.